# *Drosophila* ryanodine receptor gene triggers functional and developmental muscle properties and could be used to assess the impact of human *RYR1* mutations

**Monika Zmojdzian[1,2†], Teresa Jagla[1†], Florian Cherik[2], Magda Dubinska-Magiera[3], Marta Migocka-Patrzałek[3], Malgorzata Daczewska[3], John Rendu[4], Krzysztof Jagla[1*], Catherine Sarret[1,2*]**

[1]Institute of Genetics Reproduction and Development, INSERM U1103, CNRS UMR6293, Université Clermont Ferrand, Clermont-Ferrand, France; [2]Reference Centre for Neuromuscular Disorders, Department of Medical Genetics, Hôpital Estaing, CHU Clermont-Ferrand, Clermont-Ferrand, France; [3]Department of Animal Developmental Biology, Faculty of Biological Sciences, University of Wroclaw, Wroclaw, Poland; [4]Université Grenoble Alpes, INSERM U1216, CHU Grenoble Alpes, Grenoble Institute Neurosciences, Grenoble, France

**\*For correspondence:**
christophe.jagla@uca.fr (KJ);
csarret@chu-clermontferrand.
fr (CS)

†These authors contributed
equally to this work

**Competing interest:** The authors declare that no competing interests exist.

## eLife Assessment

This **important** paper provides novel information on the function of the Drosophila ryanodine receptor (RyR) during muscle development. The authors analyze the effects of a rare human mutation that causes myopathy that affects a conserved region of the gene. They present **compelling** evidence that this variant affects muscle function in flies. These results suggest that Drosophila can be used as a tool for screening additional variants.
[Editors' note: this paper was reviewed by Review Commons.]

**Abstract** The ryanodine receptor (RYR) genes encode evolutionarily conserved calcium release channels involved in a wide range of calcium-dependent biological processes. Here, we show that the sole *Drosophila* RYR gene (*dRyR*) functions in differentiated somatic and cardiac muscle as well as in developing embryonic myotubes. In the larval body wall muscles, dRyR protein localizes at the SR membranes, and *dRyR* knockdown adversely affects muscle contractility, suggesting its conserved role in calcium-triggered E-C coupling. After *dRyR* attenuation, sarcomere, and mitochondrial patterns are severely impaired, showing *dRyR* involvement in structural muscle properties. However, *dRyR* is also prominently expressed and functionally required in growing embryonic muscles. *dRyR* loss of function leads to myotube growth defects and thin myofiber phenotypes, while its overexpression induces myofiber splitting. Given the structural and functional conservation of *dRyR*, we used *Drosophila* to test the impact of one human *RYR1* variant of unknown significance (VUS). Larvae carrying p.Met4881Ile RYR1 VUS showed impaired mobility and altered structural muscle properties reminiscent of those seen in *dRyR* knockdown, thus indicating it is likely pathogenic. Overall, we show that *Drosophila dRyR* plays a conserved role in setting muscle contractility and structural muscle features. Our findings underline the still under-investigated role of *dRyR* as a promyogenic factor and provide a first example of the impact assessment of a human *RYR1* VUS in *Drosophila*.

## Introduction

The calcium ion ($Ca^{2+}$) concentration gradient is known to be a crucial second messenger signal in all eukaryotic cells. The calcium release channels encoded by ryanodine receptor genes are essential to maintaining correct $Ca^{2+}$ dynamics across biological membranes. RYR proteins are cellular sites of interactions with ryanodine, an alkaloid isolated from the stem wood of the plant *Ryania speciosa* (*Jenden and Fairhurst, 1969*) that was used for several decades as an insecticide. Because of its muscle-paralyzing effect in humans, ryanodine has been replaced by insect-specific derivatives that are non-toxic for mammals. Analysis of the RYR multi-domain structures and phylogenetic relations among different taxa yields a model suggesting that it evolved from inositol 1,4,5-trisphosphate receptor ($IP_3R$)-like ancestral $Ca^{2+}$ release channels. The RYR activity is strictly controlled and may respond to the presence of ions ($Ca^{2+}$, $Mg^{2+}$, and $Zn^{2+}$), proteins (calmodulin (Cam), and FK-506 binding protein (FKBP12/12.6)), and small molecules, such as ATP, caffeine, and ryanodine. The domain structure is highly conserved. For example, the SPIa kinase and ryanodine receptor (SPRY) domain shares high sequence identity between vertebrates and invertebrates and is engaged in protein-protein interactions with several protein families (*Hadiatullah et al., 2022*).

The number of RYR copies ranges across taxa. Mammalian genomes carry three RYR genes. For example, in humans, *RYR1*, *RYR2,* and *RYR3* are located on chromosomes 19q13.2, 1q43, and 15q13.3–14, respectively. Non-mammalian vertebrates, such as *Xenopus laevis* and chicken, have two *RYR* copies (*Ottini et al., 1996*) while in a zebrafish genome, there are five genes: *RYR1a, RYR1b, RYR2a, RYR2b,* and *RYR3* displaying high similarity to other vertebrate RYR genes (*Wu et al., 2011*). The greater number of RYR genes in zebrafish than in mammals and birds is thought to result from teleost-specific whole-genome duplication (*Howe et al., 2013*; *Postlethwait et al., 1998*). In the invertebrates *Drosophila melanogaster*, *Caenorhabditis elegans,* live scallop (*Placopecten magellanicus*), and lobster (*Homarus americanus*), a single RYR gene has been identified (*Hasan and Rosbash, 1992*; *Maryon et al., 1996*; *Murayama and Kurebayashi, 2011*; *Quinn et al., 1998*; *Xu et al., 2000*). However, through alternative splicing, the single invertebrate RYR gene produces several isoforms, thus increasing the diversity of the available protein pool (*De Mandal et al., 2019*; *George et al., 2007*).

It is well known that RYR genes are expressed in muscle cells and play a crucial role in muscle contraction, which results from excitation-contraction (E-C) coupling, a series of events involving the conversion of electrical stimulus to Ca-dependent mechanical response. However, RYRs are also expressed in many other animal tissues, including the central nervous system (*Klatt Shaw et al., 2018*; *Liu et al., 2005*) and are involved in housekeeping functions in the cells of the adult organism and in developmental processes (*Fill and Copello, 2002*).

*RYR1* mutations underlie an array of diseases, including muscle-impairing central core disease (CCD), one of the most frequent congenital myopathies, and malignant hyperthermia susceptibility (MHS), characterized by severe reaction to anesthetics, excessive heat, or exercise, which can be fatal. In addition, several *RYR2* mutations have also been implicated in cardiopathic catecholaminergic polymorphic ventricular tachycardia (CPVT) and arrhythmogenic right ventricular dysplasia of type 2 (ARVD2) (*Lanner, 2012*). Of the numerous mutations identified to date in RYR genes, most are missense mutations with single amino acid substitutions. However, several deletions, duplications, and frame shift mutations have also been identified (*Lanner, 2012*).

Here, we analyze the phylogenetic origin, expression and function of the *Drosophila* dRyR gene. Our data show that *dRyR* is not only required for contractile properties of differentiated somatic and cardiac *Drosophila* muscle but also influences muscle structure and plays an instructive role in muscle development. Structural muscle changes in a *dRyR* loss-of-function context suggest its implication in age-associated muscle decline, while severe developmental muscle defects observed in *dRyR* mutant embryos provide insights into early-onset RYR-related myopathies. Considering extensive structural and functional *dRyR* conservation, we generated a *Drosophila* model of one undiagnosed human *RYR1* variant mutation (*p.M4881Ile*) and found that it negatively impacted muscle structure and function, making it likely pathogenic. We report a detailed functional analysis of *dRyR*. Our findings pinpoint the under-investigated role of *dRyR* in embryonic muscle development and demonstrate that *Drosophila* could be used for assessing impacts of human *RYR1* variant mutations of unknown significance.

## Results

### *Drosophila dRyR* belongs to the RYR gene family and shows conserved somatic and cardiac muscle-associated expression and function

*dRyR* is the sole RYR gene family member in *Drosophila*. We applied the maximum likelihood method and the Kimura 2-parameter model (*Kimura, 1980*) to infer the evolutionary history of *dRyR*. Based on evolutionary distance studies, *RYR2* is the most ancient of the three human RYR genes (*Ding et al., 2017*) and shows the highest sequence homology with *Drosophila dRyR*. We compared sequences of *RYR2* from selected mammalian and non-mammalian vertebrates with the sequence of single invertebrate RYR, including fruit fly *dRyR* to illustrate its phylogenetic origin (*Figure 1A*; *Ding et al., 2017*; *Mackrill, 2012*; *McKay and Griswold, 2014*; *Takeshima et al., 1994*). Evolutionary analyses were conducted in MEGA X (*Kumar et al., 2018*; *Stecher et al., 2020*). We observed that single invertebrate RYR genes, here from *Drosophila* and *C. elegans*, clustered into a separate branch connected to the vertebrate *RYR2* branch that clustered 100% (*Figure 1A*). The vertebrate *RYR2* branch was then subdivided clonally from *Xenopus* through zebrafish, chick, and mouse to human. All clonal branches in vertebrates clustered 100% (*Figure 1A*), indicating that *RYR2* genes are closely related and most probably evolved from a single invertebrate RYR.

Expression of RYR genes in vertebrates has been extensively studied and described in various excitable cells, including skeletal and cardiac muscles and neurons, and in non-excitable cells, such as pancreatic beta cells and lymphocytes (Rossi & Sorrentino, 2002). By contrast, in spite of early works by *Hasan and Rosbash, 1992* and *Sullivan et al., 2000* no systematic analyses have yet been performed to assess the developmental expression pattern of the sole *Drosophila dRyR* gene.

We first tested the expression of *dRYR* at protein level. In differentiated body wall muscle of third instar larva (*Figure 1B and C*) *dRyR* was detected in a discrete striated pattern (*Figure 1B*), which in a zoom view revealed highly ordered punctate dRyR protein localization in close vicinity to discs large (Dlg)-positive T-tubules (*Figure 1C* and scheme in *Figure 1D*). Thus, in *Drosophila* muscle, like in vertebrates, dRyR localizes at the T-tubule interface, a sub-cellular localization consistent with its calcium release role at SR and E-C coupling function during muscle contraction.

We also assessed muscle-associated expression of *dRyR* transcripts (Fig. S1). Of ten *dRyR* transcript isoforms, all coded for proteins of similar amino acid (aa) length, ranging from 5113–5134 aa and molecular weight about 580 kDa (http://flybase.org/reports/FBgn0011286.htm). To test *dRyR* isoform expression, we applied the FISH-HCR technique (*Choi et al., 2016*) and four probes targeting alternative exons 10, 11, 22, and 23 (*Figure 1—figure supplement 1A*). In this setup we detected the expression of the A, B, F, G, H, J isoforms with the *dRyR* Ex10 probe, A, B, C, D, E, I isoforms with the Ex11 probe, A, B, C, D, E, F, G isoforms with the Ex22 probe and H, I, J isoforms with the Ex23 probe (*Figure 1—figure supplement 1A*). FISH-HCR experiments performed on differentiated third instar larval muscles revealed muscle-associated specific signals with all four probes (*Figure 1—figure supplement 1B-E*). *dRyR* transcripts were detected at the periphery of nuclei, in the sarcoplasm and in a repeated striated pattern following sarcomeres. Similar signals observed with the four probes indicated that at least the isoforms A and B that are commonly targeted by the Ex10, Ex11, and Ex22 probes and one of the isoforms H, I, or J targeted by the Ex23 probe were present in the functional body wall muscle. We could not, however, rule out the possibility that most or even all *dRyR* isoforms were expressed in the differentiated body wall muscles. Supporting this, we note that nine out of ten *dRyR* isoforms (isoform I being an exception) share the same transcription start site, suggesting that a common core promoter regulates their expression.

Detected muscle-associated expression of *dRyR* prompted us to test its involvement in muscle contraction and larva mobility. We observed that muscle-targeted attenuation of *dRyR* expression resulted in reduced muscle performance with a significantly longer time required for larvae to switch from the dorsal to the ventral position (*Figure 1E* – righting test) and a reduced number of muscle contraction waves compared to control recorded in 1 min (*Figure 1F* – motility test).

Finally, *dRyR-RNAi* larvae crawled inefficiently and were unable to move over a longer distance (*Figure 1G* – locomotor test). In parallel, overexpressing *dRyR* in larval muscles also impaired muscle function, with fewer contraction waves in 1 min compared to control and a slower locomotion (*Figure 1F and G*), while the time required to switch from the dorsal to the ventral position remained unchanged (*Figure 1E*).

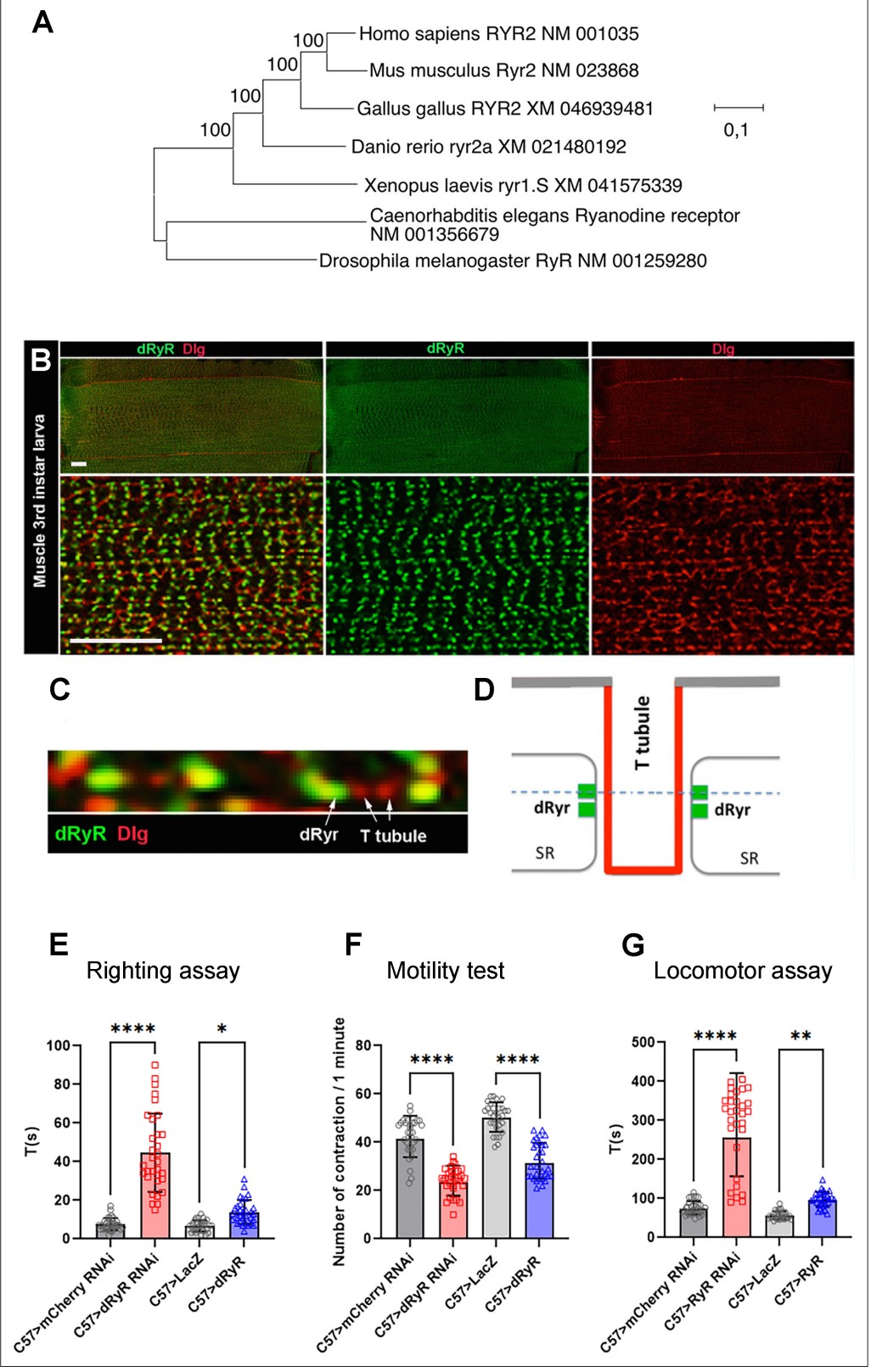

**Figure 1.** Phylogenetic origin of *Drosophila* RYR (dRyR), its body wall muscle associated expression and role in locomotion. (**A**) Evolutionary analysis by the maximum likelihood method of ryanodine receptor genes (RYR) from different taxa. The tree with the highest log likelihood (−95718.65) is shown. The percentage of trees in which the associated taxa clustered is shown next to the branches. (**B**) A wide view (upper panels) and a zoomed view

*Figure 1 continued*

(lower panels) of ventral VL3 larval muscle stained for dRyR (green) and discs large (Dlg) (red) that labels T-tubules. (**C**) A high-magnification view showing dRyR dots at the interface of T-tubules (arrows). (**D**) Scheme presenting subcellular location of dRyR receptor at the sarcoplasmic reticulum (SR) membrane in a close vicinity of T-tubules. The dotted line refers to the optical level of confocal view in (**C**). (**E–G**) Larval muscle targeted *dRyR* knockdown (C57>dRyRRNAi) leads to a marked decline in muscle performance compared to control (C57>mCherryRNAi). Three muscle performance tests were applied: (**E**) righting test, (**F**) motility test, and (**G**) locomotor test. Overexpression of *dRyR* in larval muscle (C57>dRyR) impacts muscle performance measured by the locomotor and motility tests (**F, G**). Scale bar: 20 µm. All statistical analyses were performed using Prism. The one-way ANOVA test was used for comparisons of datasets. Bar plot represent the mean and the standard deviation. On the figures, statistical comparisons of sample vs control are indicated as ****$p \leq 0.0001$; ***$p \leq 0.001$; **$p \leq 0.01$; *$p \leq 0.05$; ns>0.05.

The online version of this article includes the following source data and figure supplement(s) for figure 1:

**Source data 1.** The source data for three larval muscle performance tests: righting aasay, motility test and locomotor assay.

**Figure supplement 1.** *Drosophila* RYR (dRyR) transcript expression in larval muscles.

Altogether, our data extend previous observations of affected muscle contractility in RyR mutants (*Sullivan et al., 2000*) and suggest that *Drosophila dRyR*, like its vertebrate counterparts, ensures correct muscle function, likely acting as a sarcoplasmic reticulum (SR) calcium release channel essential for muscle contraction and E-C coupling.

Given that vertebrate RYR genes (*RYR2* in humans) also play an instrumental role in cardiac muscle function, we sought to determine whether dRyR protein could be detected in the fly heart and whether it could influence heartbeat variables. In the adult *Drosophila* heart, dRyR protein was detected predominantly in the circular muscle fibers ensuring cardiac contractions (*Figure 2A–A*), which appear structurally affected (*Figure 2C and E*) in a heart-specific *dRyR RNAi* context (Hand >dRyR RNAi). Consistent with this, heart-targeted attenuation of *dRyR* resulted in an abnormal M-mode heart profile (*Figure 2B and D*) associated with a longer heart period (*Figure 2F*) and a slow heart rate (*Figure 2G*), with significantly increased diastolic interval (*Figure 2H*). Attenuation of cardiac *dRyR* expression also led to arrhythmic heartbeat (*Figure 2J*), increased systolic diameter (*Figure 2K*) and reduced fractional shortening (*Figure 2M*). As the diastolic diameter remained unchanged, we conclude that cardiac dRyR knockdown affects cardiac performance without causing dilated cardiomyopathy. In contrast to dRyR-RNAi, increasing *dRyR* cardiac expression had only a minor influence on cardiac variables (*Figure 2F–M*) with increased systolic diameter (*Figure 2K*) but no effect on cardiac contractility (*Figure 2M*). Overall, observed *dRyR* loss-of-function adult fly heart phenotypes with a slow heart rate and increased arrhythmia correlate with impaired cardiac function in RyR mutant larvae (*Sullivan et al., 2000*). We hypothesize that dRyR RNAi-induced impairment of $Ca^{2+}$ homeostasis could contribute to cardiac aging, for which *Drosophila* is a recognized model (*Nishimura et al., 2011*).

In view of the reduced motility of third instar larva with the attenuated *dRyR* (*Figure 1E–G*), we examined whether structural properties of body wall muscles were adversely affected. We first found that muscle-targeted attenuation of *dRyR* (C57 >dRyR RNAi) led to a significantly reduced larva body length (*Figure 3B and Q*) compared to control (*Figure 3A and Q*), an observation that correlates with previously observed (*Sullivan et al., 2000*) reduced body size of $dRyR^{16}$ mutant larvae. Though to a lesser extent, the overexpression of *dRyR* in body wall muscles also impacted larva length (*Figure 3D and Q*). These changes in larva size in loss and gain of *dRyR* function correlated with a reduced longitudinal muscle length (*Figure 3E–H and R*), which in turn correlated with shortening of Kettin/D--Titin-labelled sarcomeres (*Figure 3I–L and T*) and reduced number of myonuclei (*Figure 3E–H and S*). Because RYR-mediated calcium homeostasis involves dynamic interactions between the sites of calcium release from the SR and calcium uptake by the mitochondria (*Li et al., 2025*), we examined whether in *Drosophila* dRyR loss and/or *dRyR* gain of function could adversely affect mitochondria pattern in the larval muscles. We noted that the I band-associated striated mitochondria pattern was lost in the C57 >dRyR RNAi context (*Figure 2—figure supplement 1B*) and appeared irregular when *dRyR* was overexpressed in muscles (*Figure 2—figure supplement 1D*) compared to the wild-type larval muscles (*Figure 2—figure supplement 1A, B and E*).

Both muscle-targeted attenuation and gain of *dRyR* function led not only to impaired muscle functions but also to overall reduction of muscle size and myofibrillar disarray associated with a downsizing

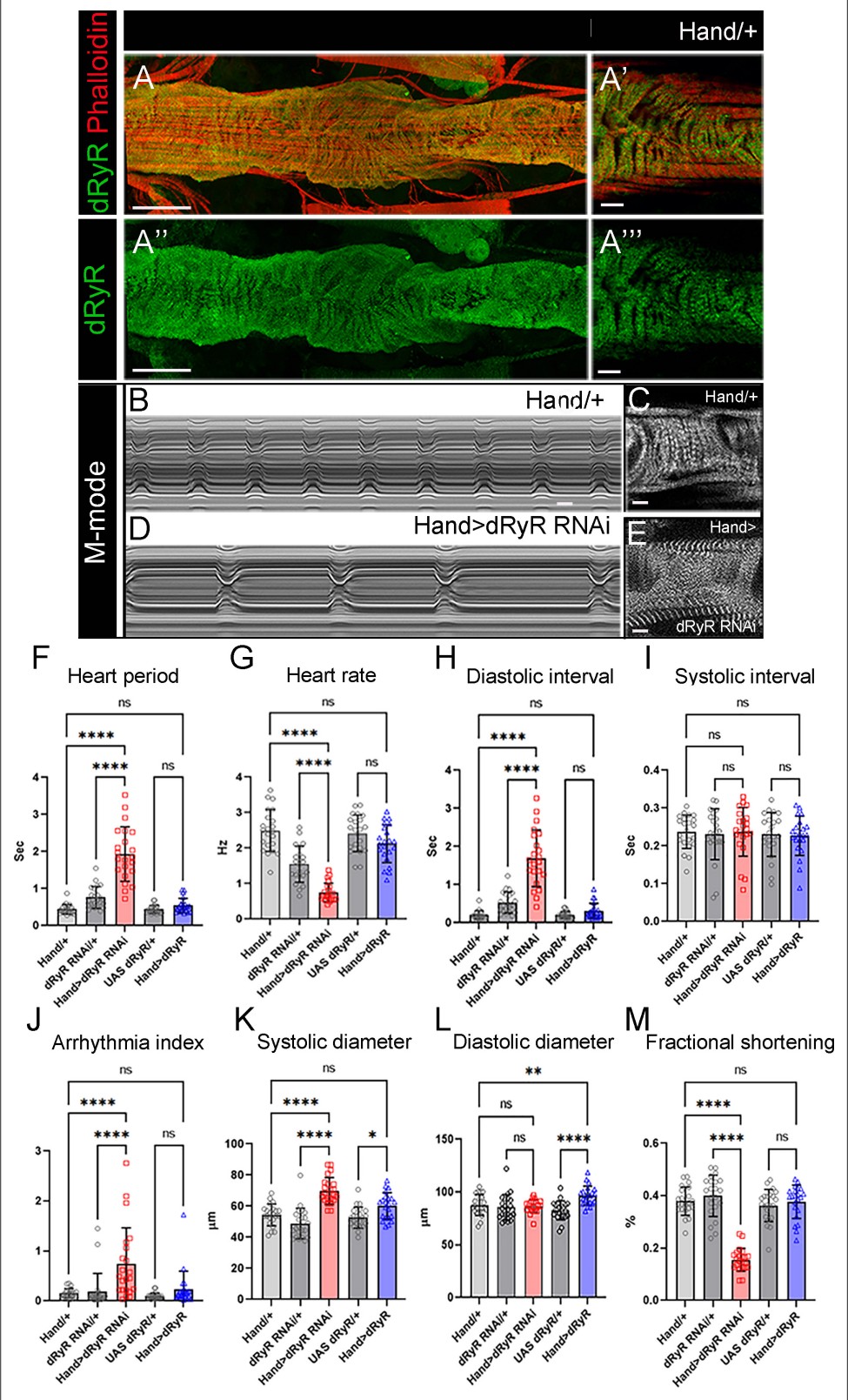

**Figure 2.** *Drosophila* RYR (dRyR) is expressed in the heart tube and is required for correct heartbeat. (**A-A'''**) Adult heart tube labeled for dRyR (green) and actin (red). (**A', A'''**) Zoomed views revealing *dRyR* expression in circular fibers. (**B,D**) M-modes of control Hand/+ (**B**) and Hand >dRyR RNAi (**D**) contexts showing a slow heart rate induced by *dRyR* attenuation. Compared with control (**C**), circular fibers in Hand >dRyR RNAi (**E**) context showed

*Figure 2 continued on next page*

*Figure 2 continued*

a fuzzy pattern suggesting an affected sarcomeric organisation. (**F–M**) Heartbeat variables in cardiac *dRyR* knockdown (Hand >dRyR RNAi) and cardiac *dRyR* overexpression contexts (Hand >dRyR). Scale bar: 50 µm in A, A″; 10 µm in A′, A‴, C, E. All statistical analyses were performed using Prism. The one-way ANOVA test was used for comparisons of datasets. Bar plot represent the mean and the standard deviation. On the figures, statistical comparisons of sample vs control are indicated as ****$p \leq 0.0001$ *$p \leq 0.05$; ns>0.05.

The online version of this article includes the following source data and figure supplement(s) for figure 2:

**Source data 1.** Source data for heartbeat variables in cardiac dRyR knockdown (Hand >dRyR RNAi) and cardiac dRyR overexpression contexts (Hand >dRyR).

**Figure supplement 1.** *Drosophila* RYR (dRyR) is required for correct mitochondria pattern.

**Figure supplement 1—source data 1.** Source data for mitochondria image analyses.

of sarcomeres and mitochondrial mismatch. Altered structural muscle features observed in *dRyR*-attenuated *Drosophila* larvae are reminiscent of myofibrillar and mitochondrial pattern defects reported in mice harboring a pathogenic *RYR1* mutation (***Elbaz et al., 2019***).

## *dRyR* is expressed during embryonic muscle development and is required for correct myogenic differentiation

Previous reports provide evidence that RYR-dependent elevation of intracellular calcium promotes late steps of myogenic differentiation and, in particular, fusion of myoblasts to myotubes (***Eigler et al., 2021***; ***Sinha et al., 2022***). In parallel, a recent study on RYR1-depleted primary myoblasts revealed the calcium-independent inhibitory role of RYR1 in myogenic differentiation (***Tourel et al., 2025***). To further explore myogenic roles of RYRs, we tested *dRyR* expression and function during embryonic development.

*Sullivan et al., 2000* reported embryonic *dRyR* transcript expression in body wall and visceral muscle precursors. Here, we tested dRyR protein expression and found that it was prominently expressed in the mesodermal derivatives in embryos. We detected dRyR protein in the developing visceral, somatic, and cardiac muscle cells (***Figure 4***). Regarding body wall muscles, dRyR could be detected in the somatic muscle precursors starting from embryonic stage 12 (***Figure 4A and A***), accumulated in the growing myotubes at mid-stage embryos (***Figure 4B and B***) and continued to be expressed in the developing muscle fibers at later embryonic stages (***Figure 4C–D***). At embryonic stage 16, dRyR protein was distributed in a discrete granular pattern within the cytoplasm of myofibers and appeared excluded from the myonuclei (***Figure 4D and D***). Thus, dRyR protein was detected from the early phase of myogenic differentiation that encompasses specification of muscle founders and first myoblast fusion events and continues during the second phase of fusion and myofiber growth and maturation. We also applied HCR-FISH to test *dRyR* transcript isoform expression in the developing somatic muscle (***Figure 3—figure supplement 1***). We found that *dRyR* A, B, F, and G isoforms harboring alternative exons 10 and 22 were actively transcribed in the developing muscle, whereas the remaining *dRyR* isoforms were barely detected (***Figure 3—figure supplement 1***).

Previous analyses (***Sullivan et al., 2000***) showed that muscle contraction was compromised in larvae carrying a hypomorphic *dRyR*[16] mutant allele. *dRyR* mutant larvae were also smaller in size and died before the pupation stage. However, whether *dRyR* embryonic expression has a functional impact on larval muscle development has not yet been assessed. Accordingly, we analyzed embryonic muscle pattern in late-stage *dRyR*[16] mutant embryos (***Figure 5***). We observed a wide range of developmental somatic muscle defects with predominant phenotype of thin myofibers present in 64% of abdominal A2-A5 hemisegments and more severely sphere-shaped (arrowheads in ***Figure 5B***) or missing myofibers (asterisks in ***Figure 5B***) observed in 25% of hemisegments. In addition, in 10% of hemisegments with *dRyR* loss of function led to supernumerary lateral transverse (LT) muscles (arrows in ***Figure 5B***), a phenotype that could arise from LT muscle splitting (***Bertin et al., 2021***). Predominant thin/misshaped/missing myofiber phenotype in a *dRyR* loss-of-function context suggests a pro-myogenic role during development. To further characterize the role of dRyR during myogenesis, we analyzed embryonic LT muscle phenotypes in LT-targeted *dRyR* attenuation and gain-of-function contexts (***Figure 5C–E***). Like in *dRyR*[16] mutant embryos, *dRyR* RNAi knockdown in LT muscles resulted in thin and/or misshaped LT muscles (***Figure 5D and H***) observed in 88% of hemisegments and in

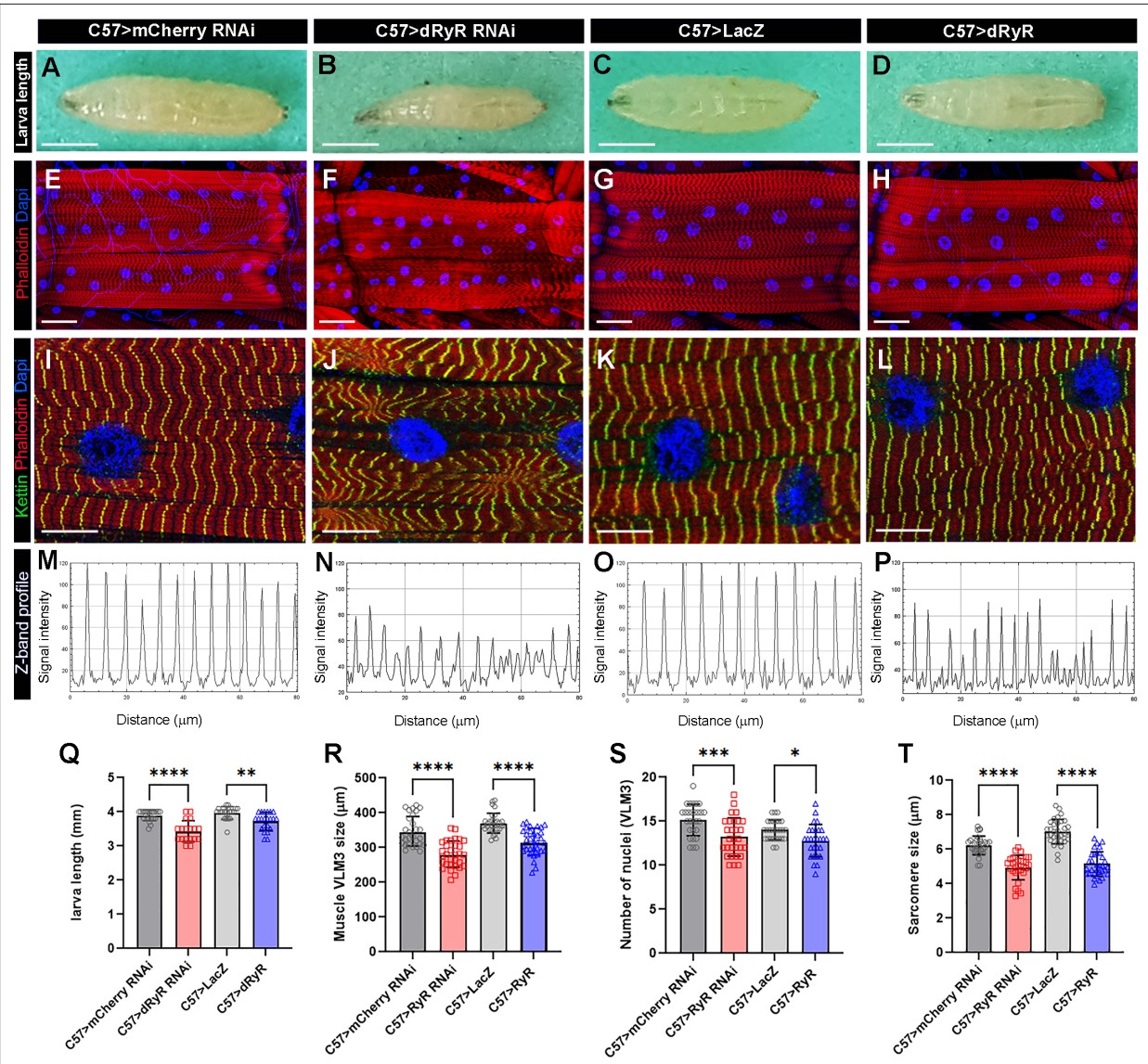

**Figure 3.** Muscle targeted *Drosophila* RYR (dRyR) loss and gain of function impacts body size and structural muscle properties. (**A–D**) General views of third instar larva in control (**A, C**), muscle-targeted *dRyR* attenuation (**B**) or overexpression (**D**). (**E–H**) Representative VL3 and VL4 ventral longitudinal muscle views from age-matched third instar larvae in control (**E, G**), C57>dRyR RNAi (**F**), and C57>dRyR (**H**) contexts. Muscle fibers and nuclei were revealed with phalloidin (red) and DAPI (blue), respectively. (**I–L**) Zoomed views of VL3 muscles of control (**I, K**), C57>dRyR RNAi (**J**), and C57>dRyR (**L**) larvae triple-stained for phalloidin (red), DAPI (blue), and Kettin/D-Titin (green). (**M–P**) Z band profiles (Kettin signal intensity plot) from zoomed views of VL3 muscles presented in (**I–L**). (**Q**) Statistical representation of third instar larva length. (**R–T**) Statistical representation of VL3 muscle characteristics: (**R**) VL3 muscle length; (**S**) number of nuclei; and (**T**) sarcomere size. Scale bar: 1 mm in A-C; 50 µm in D-F; 20 µm in G-I. Bar plots represent the mean and the standard deviation. All statistical analyses were performed using Prism. The one-way ANOVA test was used for comparisons of datasets. Bar plot represent the mean and the standard deviation. On the figures, statistical comparisons of sample vs control are indicated as ****$p \leq 0.0001$; ***$p \leq 0.001$; **$p \leq 0.01$; *$p \leq 0.05$; ns>0.05.

The online version of this article includes the following source data and figure supplement(s) for figure 3:

**Source data 1.** Source data for third instar larva length (Q) and VL3 muscle characteristics: (R) VL3 muscle length; (S) number of nuclei; and (T) sarcomere size.

**Figure supplement 1.** Embryonic expression of *Drosophila* RYR (dRyR) transcript isoforms.

rare cases of LT splitting, found in 8% of hemisegments (*Figure 5D and H*). We noted no loss of LT muscles in *dRyR* RNAi embryos. Consistent with the promyogenic role of *dRyR*, LT-targeted over-expression of *dRyR* appears to promote LT splitting phenotype (*Figure 5E and H*) found in 18% of hemisegments. Calmodulin Cam is the major calcium-dependent RYR regulator. The Ca$^{2+}$- bound Cam

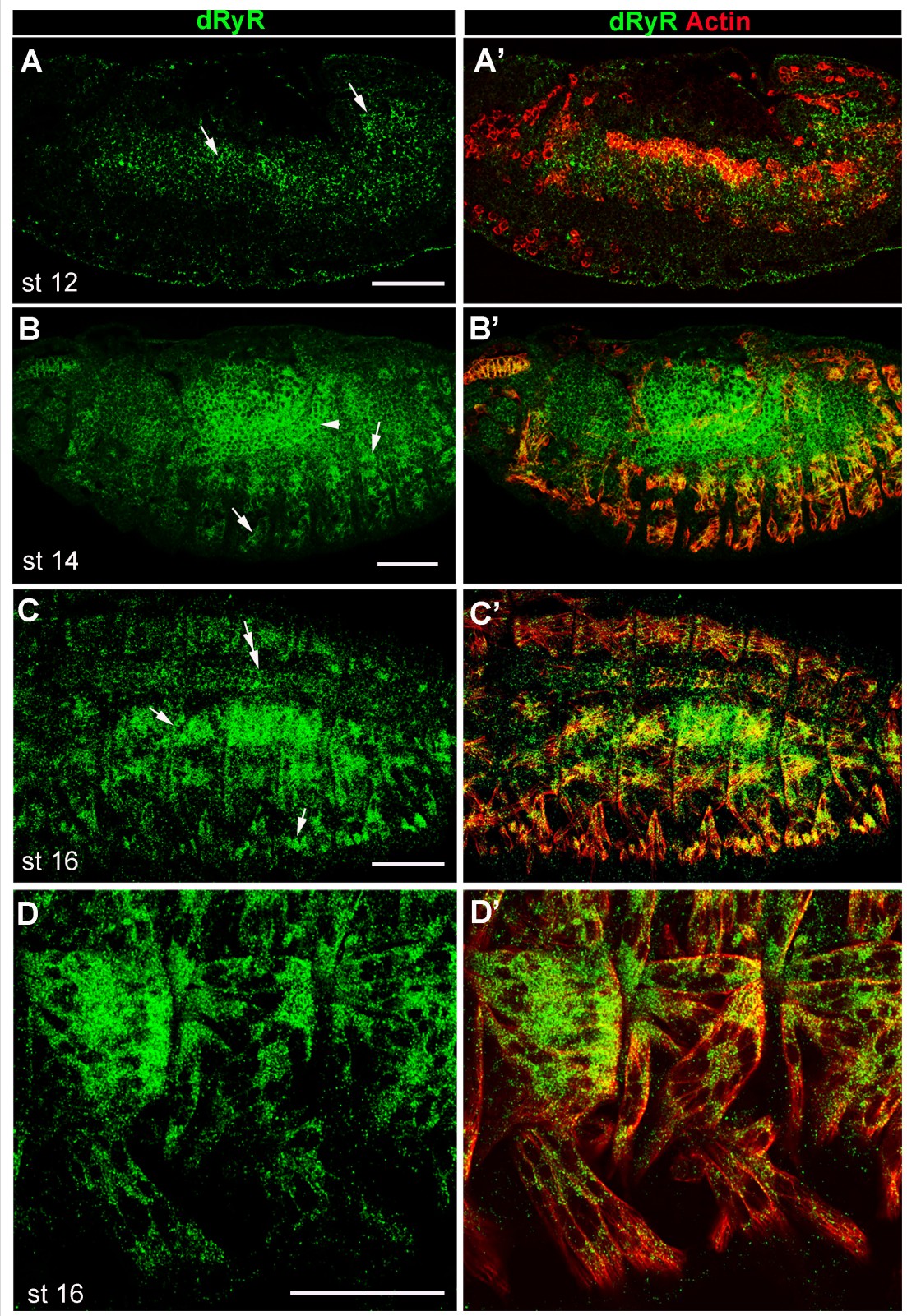

**Figure 4.** Developmental *Drosophila* RYR (dRyR) protein pattern in embryos. (**A-A'**) lateral view of a stage 12 embryo. dRyR (green) could be detected in the somatic and visceral muscle precursors (arrows in A) also revealed by Actin (red) (**A'**). (**B-C'**) dorso-lateral views of stage 14 (**B,B'**) and stage 16 (**C,C'**) embryos. dRyR accumulates in body wall muscle precursors (arrows in B and C) and in visceral muscle of the midgut (arrowhead in B) and in

*Figure 4 continued on next page*

*Figure 4 continued*
the dorsally aligned cardioblasts (double-head arrow in C). (**D,D′**) Subcellular dRyR pattern in ventral muscle precursors at embryonic stage 16. Note granular cytoplasmic distribution of dRyR. Scale bar: 50 μm.

The online version of this article includes the following figure supplement(s) for figure 4:

**Figure supplement 1.** Live imaging of developing LT muscles and GCAMP-revealed calcium levels in control and dRyR loss-of-function contexts.

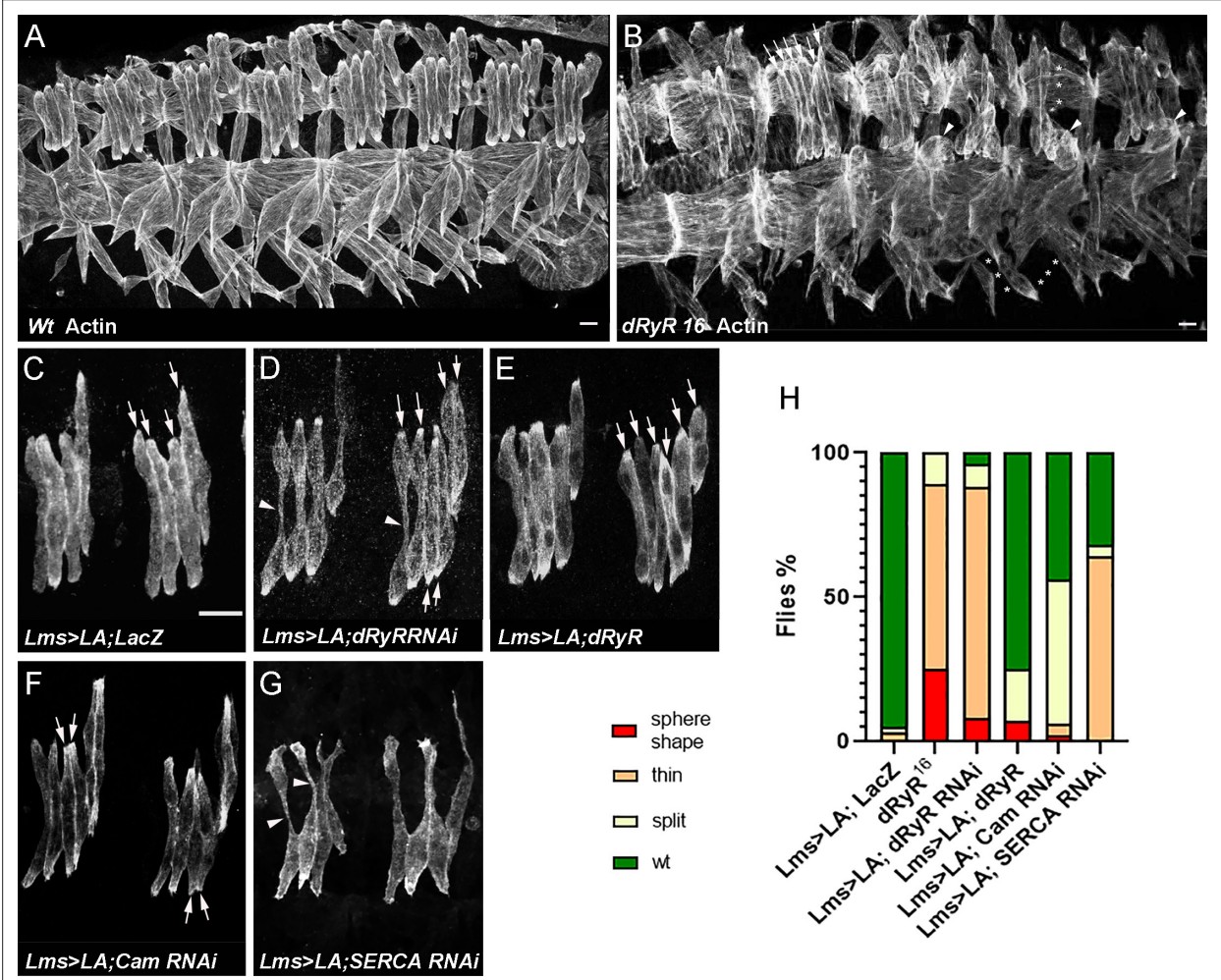

**Figure 5.** *Drosophila* RYR (dRyR) is required for correct embryonic muscle development. (**A, B**) ventro-lateral views of stage 16 embryos stained for actin to reveal embryonic muscle pattern in wild-type (**A**) and in homozygous *dRyR^16* mutant embryo (**B**). Note a wide range of developmental muscle defects that could be observed in *dRyR* loss-of-function context. Asterisks in B pinpoint muscle fiber loss, arrowheads indicate the myofibers that failed to extend and remained as myospheres and a series of arrows point to supernumerary lateral transverse myofibers (6 instead of 4). (**C–E**) Effects of lateral transverse (LT) muscle-targeted attenuation (**D**) and overexpression (**E**) of *dRyR*. Lateral transverse (LT) muscles were revealed by targeted expression of LifeActinGFP (LA) transgene using LT-specific Lms-GAL4 driver. (**C**) Four LT muscles (arrows) are seen in a control Lms >LA;LacZ context. (**D**) dRyRRNAi attenuation led to misshaped thin LTs (arrowheads) – major phenotype and to an occasional LT muscle split phenotype (6 LTs indicated by arrows). (**E**) LT targeted overexpression of *dRyR* resulted mainly in LT muscle splitting (arrows). (**F**) Cam attenuation induced mostly LT muscle splitting (arrows) while (**G**) SERCA RNAi knockdown lead to affected myofiber growth with thin LT muscle phenotype (arrowheads). (**H**) Statistical representation of LT muscle phenotypes in *dRyR* mutants and LT targeted *dRyR* knockdown, gain-of-function, and Cam and SERCA attenuation contexts. The statistical analyses were performed using Prism - contingency test; 50–60 segments/genotype. Scale bar: 10 μm.

The online version of this article includes the following source data and figure supplement(s) for figure 5:

**Source data 1.** Source dat for LT muscle phenotypes in dRyR mutants and LT targeted dRyR knockdown, gain-of-function, and Cam and SERCA attenuation contexts.

**Figure supplement 1.** Schematic comparison of amino acid sequence of human ryanodine receptor genes (RYR) and *Drosophila* dRyR proteins.

at high calcium levels acts as RYR inhibitor (*Fruen et al., 2003*). We thus tested effects of Cam attenuation in LTs and found that it results in a dRyR overexpression-like phenotype (*Figure 5F and H*). This suggests that during myogenesis Cam is present mainly in a calcium-bound form that represses dRyR. Another major regulator of calcium homeostasis, the endoplasmic reticulum calcium pump SERCA is required to maintain high calcium levels in the ER lumen (*Suisse and Treisman, 2019*). To test its role in myogenesis we analysed Lms >SERCA RNAi embryos. We observe that SERCA-depleted LT muscles display growth defects with predominant thin myofiber phenotype (*Figure 5G and H*) also observed in Lms >dRyR RNAi context (*Figure 5D and H*). Thus, both dRyR-regulated cytosolic and SERCA-regulated ER lumen calcium levels are required to promote muscle development.

Interestingly, live imaging (*Figure 4—figure supplement 1*) shows that the sphere-shaped muscle phenotype arises from the impaired LT myotube extension and not from retraction of already extended myotubes. Also, myonuclei remained on LT extremities and did not spread along the abnormally thin myofibers. The reduced number of myonuclei (2–4 per dRyR RNAi LT myofiber (*Figure 4—figure supplement 1B*, lower panel) instead of 4–6 in control LTs (*Figure 4—figure supplement 1A* lower panel)) points to the role of *dRyR* in the second wave of fusion (*Eigler et al., 2021*). This observation is consistent with the fact that overexpression of *dRyR* induced the LT split phenotype (*Figure 4E*) known to be promoted by an excessive myoblast fusion (*Bertin et al., 2021*). Because the embryonic LT muscle defects in *dRyR*-mutant embryos are associated with a reduced calcium signal in LTs (*Figure 4—figure supplement 1C*, D), we hypothesize that *dRyR* acts as a promyogenic factor ensuring correct calcium levels in the developing myotubes.

## Assessing the impact of the *RYR1* undiagnosed variant mutation in *Drosophila*

*Drosophila* dRyR shares 45% aa sequence identity with human RYR1 and RYR2, and all protein domains are conserved, with up to 75% of identity for the most C-terminal RIH domain (*Figure 5—figure supplement 1A*). Thus, although the 3D conformation of the *Drosophila* dRyR has not yet been established, high sequence and positional conservation of functional domains suggest that the conformation of dRyR protein is similar to that of its vertebrate counterparts. Also, the distribution of pathogenic mutations identified in human *RYR1* and *RYR2* genes (reviewed by *Lanner et al., 2010*), clustered in three hot spots, correlates with the positions of conserved domains. As revealed by the identity heat map (*Figure 5—figure supplement 1B*), the hot spot regions of human *RYR1 and RYR2* mutations align with the most conserved portions of the *Drosophila* dRyR indicating suitability of the *Drosophila dRyR* for modeling human *RYR* gene mutations and their impact on muscular and cardiac systems.

Over the last decades, whole genome sequencing has identified large numbers of variant mutations within the *RYR* genes, most of which are classified as variants of unknown significance (VUS). The recessive c.14643G>A/p.Met4881Ile missense *RYR1* mutation was identified in a young patient with a phenotype of congenital myopathy with a delayed acquisition of motor function. At the histopathological level, filamentous aggregates were present in muscle biopsies. This very rare mutation for which a link with a muscle disorder has not yet been evaluated, is located in the *RYR1* region encoding calcium pore.

We made use of the conserved muscle function and structural similarity between dRyR and human RYRs (*Figure 5—figure supplement 1A*) demonstrated here to generate a *Drosophila* model of variant *p.Met4881Ile RYR1* and assess its impact on larval muscle function and structure (*Figure 6*). The *p.Met4881Ile RYR1* VUS-carrying larvae were homozygous-viable but were significantly smaller (*Figure 6B and I*) than control larvae (*Figure 6A and I*). They had shorter ventral longitudinal muscles (*Figure 6D and J*), which harbored fewer myonuclei (*Figure 6D and K*) and were characterized by shorter sarcomeres (*Figure 6F and H*) compared to control (*Figure 6C, E and G*). As revealed by the larva motility tests (*Figure 6M and O*) the *p.Met4881Ile RYR1* VUS impacted not only structural but also contractile muscle properties, leading to a reduced frequency of peristaltic body wall muscle contraction (*Figure 6N*), slower larva locomotion (*Figure 6O*), and compromised muscle performance revealed by the righting test (*Figure 6M*).

These structural and functional muscle phenotypes are reminiscent of those observed in *dRyR RNAi* larvae, suggesting that *p.Met4881Ile* variant mutation negatively influences *dRyR* expression and/or function and could likely contribute to RYR-related myopathies in humans.

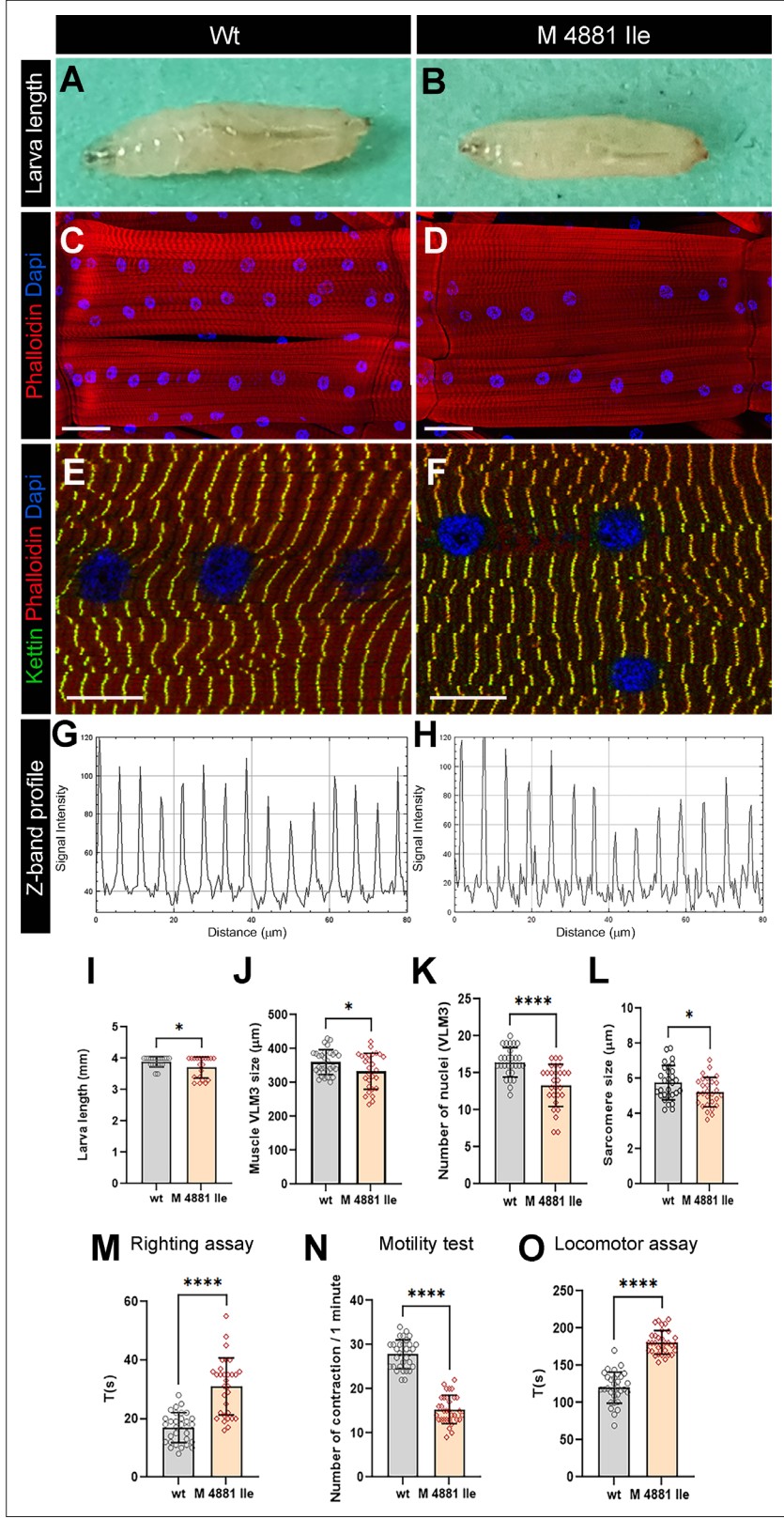

**Figure 6.** Modeling human RYR1 variant mutation in Drosophila. (**A,B**) Age-matched third instar wild-type (**A**) and *RyR1 p.Met4881Ile* mutant (**B**) larvae. Note a reduced size of larvae carrying *RYR1* variant mutation. (**C, D**) Representative views of ventral longitudinal (VL) muscles in wild-type (**C**) and *RYR1* variant mutant larvae (**D**). Note slightly reduced VL3 muscle length and reduced number of myonuclei in mutant condition. (**E–H**) Z band

*Figure 6 continued on next page*

*Figure 6 continued*

profile revealing reduction of sarcomere length in *RYR1* variant context (**F, H**) compared to control (**E, G**). Kettin is in green, Phalloidin in red, and DAPI in blue. (**I–L**) Statistical representation of larva length (**I**) and structural muscle variables (**J–L**) in wild-type and *p.Met4881Ile RYR1* variant mutation contexts. (**M–O**) Statistical assessment of functional larval muscle performance using righting test (**M**), motility test (**N**), and locomotor test (**O**) in wild-type and *RYR1* mutant conditions. Scale bar: 1 mm in A-B; 50 µm in C-D; 20 µm in E-F. All statistical analyses were performed using Prism. The *t*-test was used to compare control to variant context. Bar plot represent the mean and the standard deviation. On the figures, statistical comparisons of sample vs control are indicated as ****$p \leq 0.0001$; *$p \leq 0.05$.

The online version of this article includes the following source data for figure 6:

**Source data 1.** Source data for larva length (I), structural muscle variables (J–L) and for functional larval muscle performance using righting test (M), motility test (N), and locomotor test (O) in wild-type and RYR1 mutant conditions.

## Discussion

The ryanodine receptors encoding calcium release channels are key regulators of intracellular calcium level, largely conserved among vertebrate and invertebrate species. By their involvement in many calcium-dependent biological processes, RYR genes are vital for normal cellular functions, while mutations in RYR genes underlie a broad spectrum of human diseases, including neuromuscular and cardiac disorders (**Marks, 2023**). Because of their large size (over 500 kDa), structural complexity, many isoforms, and broad spectrum of interacting factors, even though their role in promoting muscle contraction via E-C coupling is well-characterized, other RYR functions remain only partially understood. Here, we focus on *dRyR,* a single *Drosophila* member of the RYR gene family, and characterize its expression and function in differentiated and developing muscle.

*Drosophila dRyR* shares 45% aa sequence identity with its human counterparts. Considering previous evolutionary distance studies (**Ding et al., 2017**), we show that the phylogenetic dRyR branch is connected to *RYR2*, the most ancient of the three human *RYRs*. Furthermore, a sequence alignment reveals an extensive conservation of all protein domains between *dRyR* and human RYRs, while a heatmap of conserved regions highlights previously identified hot spots of pathogenic *RYR1* and *RYR2* mutations. Remarkably, several aa residues at the functionally relevant channel pore domain (**Wang et al., 2012**) are conserved between *Drosophila* and humans.

Consistent with potential conservation of *dRyR* function in triggering $Ca^{2+}$ dynamics across ER membranes, we detected ER-associated punctate dRyR protein expression in differentiated body wall muscles. Our data extend previously reported analyses (**Sullivan et al., 2000**; **Vázquez-Martínez et al., 2003**) providing evidence that *dRyR* is not only expressed in visceral muscle and the nervous system but also prominently in differentiated striated somatic and cardiac muscle. We also wanted to know whether different *dRyR* transcript isoforms were differentially expressed in body wall muscles. However, we did not detect any such differential expression with four HCR-FISH probes targeting alternative *dRyR* exons. The fact that most *dRyR* transcript isoforms are present in differentiated larval muscles is consistent with their common transcription start sites. However, whether all *dRyR* isoforms are collectively required for larval muscle function requires further investigation. In line with muscle-associated *dRyR* expression and the excitation-contraction coupling role of its vertebrate orthologs, RNAi knockdown of *dRyR* leads to reduced muscle contractility and severely impaired larva mobility. Interestingly, in addition to impaired muscle function, *dRyR* attenuation causes extensive structural muscle defects, including reduced muscle size, smaller and aberrant sarcomeres, and degraded mitochondria pattern. All these functional and structural muscle defects are reminiscent of those of aged muscle, indicating that maintenance of *dRyR*-triggered calcium management could prevent muscle aging. This also applies to cardiac muscle, which in the *dRyR RNAi* context shows aging-associated slow heart rate and arrhythmia (**Blice-Baum et al., 2019**).

*dRyR* transcripts and dRyR protein were also detected in the embryonic muscle precursors from embryonic stage 12 to late-stage embryos, indicating that *dRyR* could be involved in managing $Ca^{2+}$ levels throughout myogenesis. In contrast to differentiated larval muscles, HCR-FISH experiments show that only a subset of *dRyR* splice isoforms is expressed in the developing muscles. Our developmental *dRyR* expression data, focusing on somatic muscle, extend previous more general analyses of *dRyR* expression and function (**Takeshima et al., 1994**; **Sullivan et al., 2000**; **Vázquez-Martínez**

*et al., 2003*). Consistent with spatiotemporal embryonic *dRyR* expression and the role of cytoplasmic calcium management (*Li et al., 2025*), we show that *dRyR* loss of function and RNAi knockdown in developing muscles cause severe developmental muscle defects. We found that *dRyR* promoted myogenic differentiation and was required for myotube growth associated with myoblast fusion and followed by myonuclear spreading within the myotubes. Our observations in *Drosophila* are consistent with the promoting role of *RYR1* in the calcium-dependent myoblast-to-myotube fusion process reported by the Avinoam lab in an in vitro myogenic differentiation system (*Eigler et al., 2021*). Interestingly, our analyses of embryonic muscle phenotypes of hypomorph *dRyR¹⁶* embryo reveal that *dRyR*, in addition to its major promyogenic role, may also negatively influence myogenic differentiation. A negative influence on myogenic differentiation and, in particular, on myoblast fusion has recently been reported in mouse *RYR1* mutant myoblast culture (*Tourel et al., 2025*). It was suggested that this early developmental role of *RYR1* was calcium-independent. *dRyR* might thus play a dual role in myogenesis: (i) as a calcium-independent negative regulator of first myoblast-to-myoblast fusion events and (ii) as a positive regulator of myogenic differentiation acting in later steps of myogenesis in a calcium-dependent way to promote myoblast-to-myotube fusion and muscle fiber growth. This major pro-myogenic *dRyR* function is further supported by the *dRyR* gain-of-function phenotypes.

The knowledge gained on *dRyR* and its muscle-associated functions prompted us to use *Drosophila* to test the impact of one human *RYR1* mutation with unknown significance. We chose undiagnosed recessive *p.Met4881Ile* variant mutation identified in a patient with congenital myopathy phenotypes, severely impaired mobility, and accumulation of filamentous aggregates in muscle fibers. We found that *Drosophila* larvae carrying *p.Met4881Ile* mutation in the *dRyR* gene showed *dRyR* RNAi-like phenotypes with impaired larval mobility and significantly impaired sarcomeric muscle structure.

In all, we assessed eight different functional and structural muscle variables showing that *p.Met4881Ile* variant mutation consistently impaired larval muscle performance and changed muscle size and structure. This suggests that the *p.Met4881Ile* mutation impairs *dRyR* function and is likely pathogenic.

To conclude, we report functional analysis of *dRyR*, the sole fruit fly *RyR* gene, and show that in addition to ensuring contractile properties of differentiated striated muscle, it plays a key pro-myogenic role during muscle development. Our findings advocate *Drosophila* for modeling and testing the impact of human *RYR1* variant mutations of unknown significance.

## Materials and methods

### Key resources table

| Reagent type (species) or resource | Designation | Source or reference | Identifiers | Additional information |
|---|---|---|---|---|
| Gene (*Drosophila melanogaster*) | RyR | | FBgn0011286 | |
| Genetic reagent (*D. melanogaster*) | p.Met4881Ile RYR1 VUS | This paper | | Generated by CRISPR-Cas9 homologous recombination genome editing Available on request |
| Genetic reagent (*D. melanogaster*) | C57-GAL4 | Bloomington *Drosophila* Stock Center | BDSC:32556; FLYB: FBti0016293; RRID:BDSC_32556 | GAL4 driver line FlyBase symbol: P{GawB}C57 |
| Genetic reagent (*D. melanogaster*) | Hand-GAL4 | Laurent Perrin, TAGC, Marseille, France | | GAL4 driver line |
| Genetic reagent (*D. melanogaster*) | Lms-GAL4 | Bloomington *Drosophila* Stock Center (unavailable) | BDSC:46861 | GAL4 driver line FlyBase symbol: P{GMR88F09-GAL4}attP2 |
| Genetic reagent (*D. melanogaster*) | RyR TRIP | Bloomington *Drosophila* Stock Center | BDSC:29445; FLYB: FBti0129073; RRID:BDSC_29445 | RNAi line FlyBase symbol: P{TRiP.JF03381}attP2 |

*Continued on next page*

*Continued*

| Reagent type (species) or resource | Designation | Source or reference | Identifiers | Additional information |
|---|---|---|---|---|
| Genetic reagent (*D. melanogaster*) | SERCA TRIP | Bloomington *Drosophila* Stock Center | BDSC_44581; FLYB: FBti0158759; RRID:BDSC_44581 | RNAi line FlyBase symbol: P{TRiP.HMS02878}attP2 |
| Genetic reagent (*D. melanogaster*) | Cam TRIP | Bloomington *Drosophila* Stock Center | BDSC:34609; FLYB: FBti0140942; RRID:BDSC_34609 | RNAi line FlyBase symbol: P{TRiP.HMS01318}attP2 |
| Genetic reagent (*D. melanogaster*) | mCherry RNAi | Bloomington *Drosophila* Stock Center | BDSC:35785; FLYB: FBti0143385; RRID:BDSC_35785 | RNAi line FlyBase symbol: P{VALIUM20-mCherry.RNAi}attP2 |
| Genetic reagent (*D. melanogaster*) | UAS-GCaMP | Bloomington *Drosophila* Stock Center | BDSC:32236; FLYB: FBti0131954; RRID:BDSC_32236 | UAS line FlyBase symbol: P{20XUAS-GCaMP3}attP2 |
| Genetic reagent (*D. melanogaster*) | UAS-RyR | Howard A Nash, University of Maryland College Park, Rockville, USA | | UAS line |
| Genetic reagent (*D. melanogaster*) | UAS-RedStinger | Bloomington *Drosophila* Stock Center | BDSC:8547; FLYB: FBti0040830; RRID:BDSC_8547 | UAS line FlyBase symbol: P{UAS-RedStinger}6 |
| Genetic reagent (*D. melanogaster*) | UAS-RedStinger | Bloomington *Drosophila* Stock Center | BDSC:8546; FLYB: FBti0040829; RRID:BDSC_8546 | UAS line FlyBase symbol: P{UAS-RedStinger}4 |
| Genetic reagent (*D. melanogaster*) | UAS-LacZ | Bloomington *Drosophila* Stock Center | BDSC:1776; FLYB: FBti0002128 RRID:BDSC_1776 | UAS line FlyBase symbol: P{UAS-lacZ.B}Bg4-1-2 |
| Genetic reagent (*D. melanogaster*) | UAS-lifeAct-GFP | Bloomington *Drosophila* Stock Center | BDSC:35544; FLYB: FBti0143326 RRID:BDSC_35544 | UAS line FlyBase symbol: P{UAS-Lifeact-GFP}VIE-260B |
| Genetic reagent (*D. melanogaster*) | RyR[16] | Bloomington *Drosophila* Stock Center | BDSC:6812; FLYB: FBal0117664 RRID:BDSC_6812 | FlyBase symbol: RyR[16] |
| Genetic reagent (*D. melanogaster*) | w[1118] | Bloomington *Drosophila* Stock Center | BDSC:3605; FLYB: FBal0117664; RRID:BDSC_3605 | FlyBase symbol: FBal0018186 |
| Antibody | anti-dlg1 (Mouse monoclonal) | Developmental Studies Hybridoma Bank (DSHB) | Cat#: 4F3 | IF(1:50) |
| Antibody | mouse anti-sls (Kettin) (Mouse monoclonal) | Developmental Studies Hybridoma Bank (DSHB) | Cat#: 1B8-3D9 | IF(1:50) |
| Antibody | anti-GFP (Goat polyclonal) | Abcam | Cat#: ab5450 | IF(1:500) |
| Antibody | anti-ATP5A (Mouse monoclonal) | Abcam | Cat#: ab14748 | IF(1:200) |
| Antibody | anti-Actin (Rat monoclonal) | Abcam | Cat#: ab50591 | IF(1:500) |
| Antibody | anti-beta galactosidase (Chicken polyclonal) | Abcam | Cat#: ab9361 | IF(1:1000) |

*Continued on next page*

*Continued*

| Reagent type (species) or resource | Designation | Source or reference | Identifiers | Additional information |
|---|---|---|---|---|
| Recombinant DNA reagent | PCFD5 plasmid | Adgene | Plasmid #73914 | |
| Sequence-based reagent | RyR F | This paper | PCR primers | 5'-TGCAGAGCAGCCGGAGGATGAC |
| Sequence-based reagent | RyR R | This paper | PCR primers | 5'-ATCAGACGCGGCGAATCCGCAG |

### *Drosophila* strains and genetics

Fly stocks were maintained at 25 °C on standard fly food.

The targeted expression experiments were performed using the UAS-GAL4 system (*Brand and Perrimon, 1993*) on the following GAL4 and UAS lines: C57-GAL4 (Bl32556); Hand-GAL4 (kindly provided by L. Perrin; TAGC, Aix-Marseille University, France); UAS-dRyR RNAi (BL29445); Lms-GAL4 (BL46861); UAS-mCherry RNAi (BL35785); UAS-dRyR (kindly provided by H. Nash University of Maryland College Park, Rockville, USA), UAS-lifeAct-GFP (BL35544), UAS-dsRed NLS (BL8547; BL8546), UAS-LacZ (BL1776), UAS-Cam RNAi (Bl34609), UAS-SERCA RNAi (BL44581), and UAS-GCaMP3 (BL32236). The RyR[16]/CyO Wg LacZ (BL6812) was used as hypomorphic mutant and the $w^{1118}$ strain was used as wild-type.

### Phylogenetic analysis

Initial trees for the heuristic search were obtained automatically by applying Neighbor-Join and BioNJ algorithms to a matrix of pairwise distances estimated using the maximum composite likelihood (MCL) approach and then selecting the topology with superior log likelihood value (*Kumar et al., 2018*). The tree is drawn to scale, with branch lengths measured in number of substitutions per site. This analysis involved seven nucleotide sequences. Codon positions included were first + second + third + Noncoding. There were 17,322 positions in the final dataset.

### Immunohistochemistry

Antibody staining was performed using standard protocol. Embryos were fixed in 4% formaldehyde and blocked in NGS serum to remove non-specific epitopes. They were incubated overnight at 4 °C with primary antibodies followed by secondary antibodies for 2 hr at RT.

Third instar larvae were dissected and fixed in 4% paraformaldehyde for 20 min as previously described (*Lavergne et al., 2020*). The fly hearts were dissected (*Fink et al., 2009*; *Ocorr et al., 2007*) and fixed for 15 min in 4% formaldehyde and the immunostaining procedure was performed as described (*Auxerre-Plantié et al., 2019*).

The following primary antibodies were used in this study: guinea pig anti-dRYR antibody (1:200, kindly provided by Robert Scott and Benjamin White from NIH/NIMH Institute and previously described in *Gao et al., 2013*), mouse anti-Dlg (1–50, DSHB, 4F3), rat anti-actin (1–500, Abcam, ab 50591), mouse anti-kettin (1:50 DSHB 1B8-3D9), mouse anti-ATP5A (1:200 Abcam, ab 14748), goat anti-GFP (1:500, Abcam, ab 5450), and chicken anti-β galactosidase (1:1000, Abcam, ab 9361). Rhodamine phalloidin (Thermo Fischer Scientific) was used to reveal actin filaments in the heart and muscles. Fluorescent secondary antibodies (Jackson ImmunoResearch) were used to detect primary antibodies.

### Muscle characteristics measurements

All analyses of muscle length and sarcomere size were performed on fixed larval muscle preparations in a relaxed state. Acquired confocal images were analysed in Fiji using the *line tool*. *Analyze – Measure* tool was then applied to obtain muscle length values and measurements were analysed with Prism. Sarcomere size and number were calculated using *Analyze – Plot profile* Fiji tool. The sarcomere size was measured between peaks corresponding to Z-disc (revealed with Z-line specific marker) on approximately 100 µm of muscle length. Sarcomere measurements were then analysed with Prism.

DAPI-stained nuclei were counted in Z-stacks of confocal views of VL3 larval muscle and data analyzed with Prism. About 30 larval muscles from 6 to 8 larval filets were analysed for each measurement.

## In situ hybridization chain reaction – HCR

In this study, we used the two-stage in situ HCR protocol described by *Choi et al., 2016*. This technique detects and amplifies specific transcripts by the direct binding of probes to nucleic target sequences without additional long-lasting enzymatic reaction. We planned four different mRNA probes targeting alternative exons numbered 10, 11, 22, and 23.

We used a Molecular Instruments HCR kit containing a DNA probe set, a DNA HCR amplifier B1-Alexa fluor 488, B2-fluor 532 and hybridization, wash and amplification buffers.

Fixed samples were pre-hybridized at 65 °C for 2 hr followed by hybridization steps overnight at 45 °C. After several washes, the amplification step was performed overnight at RT in the dark. We used TRJ244 HCR amplifier B1, RTJ245 HCR amplifier B2, RTJ256 HCR amplifier B2, RTJ 247 HCR amplifier B1. Target mRNA sequence information remains at the discretion of Molecular Instruments Company. Excess of hairpins was removed by several washes with 5XSSCT solution at room temperature.

## Imaging

Samples were mounted in Vectashield with DAPI (Vector Laboratories, Inc Burlingame, CA) and a Leica SP8 confocal microscope was used for image acquisition and for time-lapse imaging of living embryos. In vivo imaging of lateral muscle was performed from late stage 14 to stage 16 every 3 min. We used Adobe Photoshop and ImageJ for image processing.

## Genome editing

To generate the *Drosophila* model of human *RYR1* VUS p.Met4881Ile we applied the CRISPR-Cas9 homologous recombination genome editing approach with the use of single-strand oligo donor (ssODN) and gRNA. A 20nt 5'-AAACGCTTCGTGTTCCATCTGTAC-3' gRNA targeting the *dRYR* region close to the mutation site was designed using the OPTIMAL Target Finder platform (*Gratz et al., 2014*) and cloned into PCFD5 plasmid (Adgene). Single-strand ssODN donor of 115 nt in size with sequence modification TTG to ATC was generated by IDT Company. Both components were injected at a concentration of 100 ng/μl by Best Gene into nos-Cas 9 (III-attp2) flies. PCR molecular screening with a pair of primers: Forward 5'-TGCAGAGCAGCCGGAGGATGAC; Reverse 5'-ATCAGACGCGGCGAATCCGCAG and Sanger sequencing were used to identify progenies carrying the edited sequence. Genetic crosses established homozygous *Drosophila* lines carrying the RyR1 p.Met4881Ile variant mutation.

## Functional tests of muscle performance

Motility tests were carried out on 30 third instar larvae as previously described (*Picchio et al., 2013*). The motility test was performed by recording the number of peristaltic contractions executed by the larva in 1 min on a grape medium plate. The righting test consisted in placing the larva on its back and measuring the time it took to revert to its crawling position. For the locomotor test, a track 2 mm wide, 5 mm deep and 5 cm long was prepared on a grape medium plate. Larvae were placed on the test track and the time they took to crawl a distance of 5 cm was recorded.

## Heartbeat analyses

The cardiac activity analyses of adult *Drosophila* hearts were performed on 1-week-old female flies using the Semi-automated Optical Heartbeat Analysis (SOHA) approach protocol of *Fink et al., 2009*. For each experiment, about 20 flies were analyzed. The flies were anesthetized with Fly Nap, dissected in an oxygenated, artificial hemolymph composed of 108 mM NaCl, 5 mM KCl, 2 mM CaCl$_2$, 8 mM MgCl$_2$, 1 mM NaH$_2$PO$_4$, 4 mM NaHCO$_3$, 10 mM sucrose, 5 mM trehalose, 5 mM HEPES (pH 7.1). The beating hearts were filmed by digital camera on 30 s movie with the speed of 150 frames/s (Digital camera C9300, Hamamatsu, McBain Instruments, Chatsworth, CA). The SOHA program, based on Matlab R2009b software, was used for film analysis (*Fink et al., 2009*).

## Mitochondria area quantifications

The total area of mitochondria was identified by Otsu thresholding on Fiji for internal muscle z-planes. The quantifications for z-planes were averaged for each of 25–30 different VL3 muscles according to *Zhang et al., 2024*.

## Statistic

All statistical analyses were performed using Prism (v9.5.1, GraphPad Software, La Jolla, CA, USA). The *t*-test was used to compare control to variant context and one-way ANOVA tests were used for comparisons with more than two datasets. Bar plot represents the mean and the standard deviation. On the figures, statistical comparisons of sample vs control are indicated as ****$p \leq 0.0001$; ***$p \leq 0.001$; **$p \leq 0.01$; *$p \leq 0.05$; ns>0.05.

## Acknowledgements

We thank Drosophila Bloomington Stock Centers for Drosophila lines and L Mouty for the technical assistance. This study was supported by the 'Priority Research Programme on Rare Diseases' of the French Investments for the Future Programme, by the AFM-Telethon strategic grant to MyoNeurAlp consortium, the Reference Centre for Neuromuscular Diseases grant to MZ and CS, ANR-iSITE-2025 grant to MZ, by the French Government Scholarship grant to MMP and The Polish National Agency for Academic Exchange within The Bekker Programme (grant no. BPN/BEK/2021/2/00006) to MDM.

---

## Additional information

### Funding

| Funder | Grant reference number | Author |
|---|---|---|
| Institut National de la Santé et de la Recherche Médicale | RYR-ClassifAI | John Rendu Krzysztof Jagla |
| Department of Agriculture, Food and the Marine, Ireland | MyoNeuralp | Monika Zmojdzian |
| Agence Nationale de la Recherche | iSITE CAP2025 | Krzysztof Jagla Catherine Sarret |
| Bekker Programme | BPN/BEK/2021/2/00006 | Magda Dubinska-Magiera Marta Migocka-Patrzałek |

The funders had no role in study design, data collection and interpretation, or the decision to submit the work for publication.

### Author contributions

Monika Zmojdzian, Teresa Jagla, Formal analysis, Investigation, Methodology, Writing – original draft; Florian Cherik, Investigation, Methodology; Magda Dubinska-Magiera, Formal analysis, Investigation, Writing – original draft; Marta Migocka-Patrzałek, Investigation, Methodology, Writing – original draft; Malgorzata Daczewska, Validation, Methodology; John Rendu, Conceptualization, Resources, Funding acquisition, Methodology; Krzysztof Jagla, Conceptualization, Formal analysis, Supervision, Funding acquisition, Writing – review and editing; Catherine Sarret, Supervision, Methodology, Writing – review and editing

### Author ORCIDs

Monika Zmojdzian ⬤ https://orcid.org/0000-0001-6174-2719
Teresa Jagla ⬤ https://orcid.org/0000-0002-9277-6089
Marta Migocka-Patrzałek ⬤ https://orcid.org/0000-0003-0077-7779
John Rendu ⬤ https://orcid.org/0000-0002-0377-0807
Krzysztof Jagla ⬤ https://orcid.org/0000-0003-4965-8818

## Ethics

This study was conducted in accordance with the ethical standards of the institutional and national research committees. Ethical approval was obtained from the appropriate institutional ethics committee under the reference 38RC21.0399.

Reviewer #1 (Public review): https://doi.org/10.7554/eLife.111053.2.sa1
Reviewer #2 (Public review): https://doi.org/10.7554/eLife.111053.2.sa2
Author response https://doi.org/10.7554/eLife.111053.2.sa3

---

# Additional files

## Supplementary files
MDAR checklist

## Data availability
All behavioral, phenotypical and image analyses datasets that were used to generate graphs in Figures 1–6 are available as source data files.

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
