## [Editor Report · eLife Assessment]

This **important** paper provides novel information on the function of the Drosophila ryanodine receptor (RyR) during muscle development. The authors analyze the effects of a rare human mutation that causes myopathy that affects a conserved region of the gene. They present **compelling** evidence that this variant affects muscle function in flies. These results suggest that Drosophila can be used as a tool for screening additional variants.

[Editors' note: this paper was reviewed by Review Commons.]

---

## [Referee Report · Reviewer #1 (Public review)]

Summary:

In this manuscript, Zmojdzian et al. provide an analysis of ryanodine receptor (RyR) expression and function in Drosophila. They also use CRISPR to engineer into flies a RyR variant of unknown significance (VUS) found in a human myopathy patient and demonstrate that it is likely a pathogenic mutation. From studies of RyR expression in embryonic and larval stages, and effects of RyR knockdown or overexpression in various muscle groups, the authors show that, in addition to its known actions in calcium-dependent excitation-contraction coupling, RyR promotes myogenesis during development.

The key conclusions of the paper are convincing. I do not have suggestions for necessary additional experimental work, and my comments are minor. One conclusion, that RyR dysfunction may be involved in aging, is stated in multiple places, sometimes speculatively but once very forcefully. The latter is in the final paragraph of the Discussion, which states RyR "plays an instrumental anti-aging role in differentiated striated muscle". This conclusion must be tempered, as even if RyR knockdown phenotypes resemble some of those seen in aging flies, the study does not examine aged flies, and there is no mechanistic analysis that might link the two. I assume the authors would prefer to modify that sentence than initiate work with aging flies to prove the assertion. Finally, the use of CRISPR to test a VUS is excellent and suggests a good way for testing of additional RyR variants in the future.

Significance:

The paper is significant in that RyR is known to be a critical protein in calcium-dependent excitation-contraction coupling but its role in developmental myogenesis is poorly studied. This study demonstrates that it is expressed during, and is important for, embryonic and larval myogenesis in the fly. RyR is also understudied in this valuable model organism, even though a P element-based mutant has been available since 2000. The mechanistic basis for the functional observations is not explored here but the work is well performed and will be of interest to investigators studying muscle development (my own field) and diseases caused by RyR mutations.

---

## [Referee Report · Reviewer #2 (Public review)]

Summary:

This paper presents data using the Drosophila model to analyze the effects of a rare human mutation in the gene encoding the ryanodine receptor (ryr). The authors present a nice, comprehensive phylogenetic analysis that shows the Drosophila version of Ryr to be most similar to human RYR2 and that the known "hot spots" for mutations in RYR2 coincide with highly conserved regions of the Drosophila Ryr. They characterize the functional effects of ryr knockdown and overexpression on both adult heart function and larval body wall muscle. They identified embryonic ryr expression in association with actin-stained muscle precursor cells and provide beautiful stains, which clearly showed that embryonic muscle cell development was disrupted in ryr mutants. In support of these findings, KD of Calmodulin in larva (an Ryr inhibitor) phenocopied Ryr OE. They recreated a human variant of unknown function (RyR1 p.Met4881Ile) in the conserved region of the fly gene and tested the effect on larval muscle. Their data suggested that this variant was likely deleterious as it negatively affected most muscle parameters.

Major comments:

(1) Fig, 1 In G there is no data for the RNAi KD situation.

(2) Fig. 2 Authors should include Diastolic Diameters; they mention dilated cardiomyopathy but don't show the dilation. The authors should also show staining in hearts with RYR OE and RNAi. It would be nice to have some kind of quantification of disorganized myofibrils.

(3) To evaluate and reproduce the data on the larva muscle parameters the authors should provide more details on how sarcomere length was quantified in each larva (replicates, ROI size, etc). Similarly, how were # of nuclei quantified / normalized? Importantly for these measurements, did the authors know what the contraction state of the muscles were when fixed?

(4) Fig. 3, Are RNAi and OE in the same background? I only see one control in the graphs for the RNAi line background.

(5) Fig. 3 How VL3 length was determined needs more detail, the Zhang ref is not adequate.

(6) In order to be able to evaluate the data, the statistical tests used should be cited in the figure legends along with what *, ** ,*** stand for (or just provide p values).

Significance:

The authors nicely characterized the role of Ryr in muscle development and function and recreated a human variant of unknown function (RyR1 p.Met4881Ile) in the conserved region of the fly gene. Their data suggested that this variant was likely deleterious as it negatively affected most muscle parameters. This work supports a role for the fly model in testing potential human disease gene variants.

Comments on Revised Version:

The authors have very adequately addressed the points raised by all reviewers.

---

## [Author Response]

**General Statements**

We would like to extend our gratitude to all reviewers for their supportive feedback, which acknowledges our study as well performed and of interest to investigators studying muscle development and diseases and supporting a role for the fly model in testing potential human disease gene variants. We also thank the reviewers for their valuable critical comments. We carefully considered all of them and made additional experiments and suggested text amendments.

We believe these modifications substantially improve the quality of our results and enhance general interest of our work.

**Point-by-point description of the revisions**

**Reviewer #1:**
In this manuscript, Zmojdzian et al. provide an analysis of ryanodine receptor (RyR) expression and function in Drosophila. They also use CRISPR to engineer into flies a RyR variant of unknown significance (VUS) found in a human myopathy patient and demonstrate that it is likely a pathogenic mutation. From studies of RyR expression in embryonic and larval stages, and effects of RyR knockdown or overexpression in various muscle groups, the authors show that, in addition to its known actions in calcium-dependent excitation-contraction coupling, RyR promotes myogenesis during development.The key conclusions of the paper are convincing. I do not have suggestions for necessary additional experimental work, and my comments are minor. One conclusion, that RyR dysfunction may be involved in aging, is stated in multiple places, sometimes speculatively but once very forcefully. The latter is in the final paragraph of the Discussion, which states RyR "plays an instrumental anti-aging role in differentiated striated muscle". This conclusion must be tempered, as even if RyR knockdown phenotypes resemble some of those seen in aging flies, the study does not examine aged flies, and there is no mechanistic analysis that might link the two. I assume the authors would prefer to modify that sentence than initiate work with aging flies to prove the assertion.

We thank the Reviewer for this comment and remove from the concluding sentence hypothetical anti-aging role of RyR. The modified sentence reads as follow:

“To conclude, we report functional analysis of dRyR, the sole fruit fly RyR gene and show that in addition to ensuring contractile properties of differentiated striated muscle it plays a key pro-myogenic role during muscle development.”

Finally, the use of CRISPR to test a VUS is excellent and suggests a good way for testing of additional RyR variants in the future.Minor comments:(1) Figure 1A: In the Introduction it is stated that non-mammalian vertebrates have two RyR genes, alpha and beta. In Fig. 1A, a single chicken and single frog gene are listed under names different than alpha or beta. The figure also focuses on RyR2 genes, yet the Introduction states that the non-mammalian vertebrate genes are homologous to RyR1 and RyR3 in mammals. The dichotomy between the text and the figure is confusing. Finally, the font used in Fig. 1A should be enlarged for better visibility.

To avoid the dichotomy we modified our sentence concerning the non-mammalian vertebrate RYR genes in the Introduction section. As indicated, there are two RYR genes in chicken and frog, with one that shares homology with vertebrate RYR2 and is represented in the phylogenetic tree (Fig. 1A). As requested by the reviewer, to ensure better visibility we enlarged the font in the revised Fig. 1A.

(2) Figure 3G-I: IF to Kettin is used to reveal sarcomeres but is not mentioned in the text. This protein is not present in vertebrates (I believe) and may not be familiar to many readers. It should be described in the text when it is used.

We are grateful for reminding us to provide information about Kettin, which represents the Drosophila counterpart of Titin. The following information has been added to the text on page 9: “ …which in turn correlated with shortening of Kettin/D-Titin-labelled sarcomeres…”

(3) Figure S2: The panels are labelled E, F, G. They should be A-D, as is used in the text.

In the revised version of Fig. S2 panel labels were amended and the panel E view enlarged. We also provide an additional control context (C57>LacZ).

(4) The dRyR16 allele is used in Figure 5 and S4. It is described as a hypomorph in the text on page 12 but as a null in the legend to Figure 5. Do the authors actually mean "homozygous" in the legend? The difference should be clarified.

The dRyR^16^ allele has been previously described as hypomorph. Indeed, in the legend of Fig. 5 we by mistake describe it as a “null”. As suggested by the Reviewer we modify it to « homozygous ».

(5) The Met codon that is mutated in the variant studied in Figure S5 and Figure 6 is position 488 in humans. It is referred to that way in the fly version also. Is that true, the actual amino acid number is identical in humans and flies? In Figure S5B, it might be worth showing the primary amino acid sequence surrounding Met488 to reveal the degree of local conservation (beyond the orange domain in that panel).

To provide more information about the conservation we include to the revised Fig. S5 an alignment of amino acid sequence surrounding the human RYR1 4881 variant position, which corresponds to position 4971 in the Drosophila dRyR.

Author response image 1 shows a snapshot from a larger portion of alignment encompassing variant mutation showing a high amino acids conservation around the variant position:

(6) At least two references cited in the text are not listed in the References section (Hadiatullah et al. and Nishimura et al.).

We double check reference citation and two indicated positions are now listed in the References section.

**Reviewer #1 (Significance):**
The paper is significant in that RyR is known to be a critical protein in calcium-dependent excitationcontraction coupling but its role in developmental myogenesis is poorly studied. This study demonstrates that it is expressed during, and is important for, embryonic and larval myogenesis in the fly. RyR is also understudied in this valuable model organism, even though a P element-based mutant has been available since 2000. The mechanistic basis for the functional observations is not explored here but the work is well performed and will be of interest to investigators studying muscle development (my own field) and diseases caused by RyR mutations.

To reinforce mechanistic/functional side of our studies we include to the revised Fig.5 a new panel G showing promyogenic role of another major cellular calcium regulator, ER calcium pump SERCA. The Lms targeted RNAi knockdown of SERCA leads to affected myotube growth resulting in a thin muscle fiber phenotype. This indicates that both dRyR-regulated cytosolic and SERCA-regulated ER store calcium levels are required to promote muscle development.

**Reviewer #2:**
Summary:This paper presents data using the Drosophila model to analyze the effects of a rare human mutation in the gene encoding the ryanodine receptor (ryr). The authors present a nice, comprehensive phylogenetic analysis that shows the Drosophila version of Ryr to be most similar to human RYR2 and that the known "hot spots" for mutations in RYR2 coincide with highly conserved regions of the Drosophila Ryr. They characterize the functional effects of ryr knockdown and overexpression on both adult heart function and larval body wall muscle. They identified embryonic ryr expression in association with actin-stained muscle precursor cells and provide beautiful stains, which clearly showed that embryonic muscle cell development was disrupted in ryr mutants. In support of these findings, KD of Calmodulin in larva (an Ryr inhibitor) phenocopied Ryr OE. They recreated a human variant of unknown function (RyR1 p.Met4881Ile) in the conserved region of the fly gene and tested the effect on larval muscle. Their data suggested that this variant was likely deleterious as it negatively affected most muscle parameters. This work supports a role for the fly model in testing potential human disease gene variants.Major comments:(1) Fig, 1 In G there is no data for the RNAi KD situation.

We are grateful to the Reviewer for pointing this out. We initially didn’t include these data because of large difference in crawling capacities of dRyR RNAi larvae. In the revised version of Fig. 1 we provide now dRyR-RNAi larva crawling data. Because of their inefficient crawling, the time scale in panel 1G was modified.

(2) Fig. 2 Authors should include Diastolic Diameters; they mention dilated cardiomyopathy but don't show the dilation. The authors should also show staining in hearts with RYR OE and RNAi. It would be nice to have some kind of quantification of disorganized myofibrils.

As requested, in the revised Fig. 2 we provide diastolic diameter measures. We also include systolic interval graph to show a full picture of cardiac parameters. We do not observe all signs of dilated cardiomyopathy in dRyR-RNAi context as there is systolic diameter increase but no significant change in diastolic diameter.

We modify our comments in the text accordingly (page 7).

“…As the diastolic diameter remained unchanged, we conclude that cardiac dRyR knockdown affects cardiac performance without causing dilated cardiomyopathy…”

Regarding circular myofibrils pattern, we do not observe irregularity of myofibrils orientation but rather a fuzzy and less distinctive sarcomeric pattern that is difficult to quantify. We specify this in the figure 2 legend (page 8).

“…circular fibers in Hand>dRyR RNAi (E) context showed a fuzzy pattern suggesting an affected sarcomeric organisation…”

Author response image 2 shows the entire view of the cardiac tube in dRyRRNAi context (stained with phalloidin) in which in spite of less distinctive circular myofibrils no obvious differences with wt are observed.

**Author response image 2. sa3fig2:** 

(3) To evaluate and reproduce the data on the larva muscle parameters the authors should provide more details on how sarcomere length was quantified in each larva (replicates, ROI size, etc). Similarly, how were # of nuclei quantified / normalized? Importantly for these measurements, did the authors know what the contraction state of the muscles were when fixed?

We add the requested information to the Materials and Methods section:

“Muscle characteristics measurements:

All analyses of muscle length and sarcomere size were performed on fixed larval muscle preparations in a relaxed state. Acquired confocal images were analysed in Fiji using the line tool. Analyze – Measure tool was then applied to obtain muscle length values and measurements were analysed with Prism. Sarcomere size and number were calculated using Analyze – Plot profile Fiji tool. The sarcomere size was measured between peaks corresponding to Z-disc (revealed with Z-line specific marker) on approximatively 100 µm of muscle length. Sarcomere measurements were then analysed with Prism.

DAPI-stained nuclei were counted in Z-stacks of confocal views of VL3 larval muscle and data analysed with Prism. About 30 larval muscles from 6-8 larval filets were analysed for each measurement. » Statistics

All statistical analyses were performed using Prism (v9.5.1, GraphPad, Software, La Jolla, CA, USA). The t test was used to compare control to variant context and one-way ANOVA tests were used for comparisons with more than two datasets. Bar plot represent the mean and the standard deviation. On the figures, statistical comparisons of sample vs control are indicated as ****: P ≤ 0.0001; ***: P ≤ 0.001; **: P ≤ 0.01; *: P ≤ 0.05; ns > 0.05.

(4) Fig. 3, Are RNAi and OE in the same background? I only see one control in the graphs for the RNAi line background.

We agree and to avoid potential bias between the RNAi versus OE genetic contexts we provide now in the revised version of Fig. 3 an additional OE control (C57>lacZ).

Thus, two controls, one for RNAi and one for OE contexts are now included.

(5) Fig. 3 How VL3 length was determined needs more detail, the Zhang ref is not adequate.

We are thanking the Reviewer for this comment and provide now more details about the method used to calculate VL3 length (new paragraph in Materials and Methods), see also our answer to point 3. Zhang et al. reference is in relation to the mitochondria pattern quantification.

(6) In order to be able to evaluate the data, the statistical tests used should be cited in the figure legends along with what *, ** ,*** stand for (or just provide p values).

We add now the information about the statistical tests to the Fig legends in addition to the specific paragraph in Materials and Methods section (answer to point 3).

Minor comments:(1) Need more detail in the figures, e.g. add what colors go with which stain to the picture.

We provide this information in the revised version of the figure legends

(2) Page 13, (Fig. ?F, G).

We apologize for this mistake and add the number - Fig. 5

(3) Fig. 4 "partially co-localizing with actin".... this is confusing and probably an overstatement based on the staining pattern in a whole embryo and not on an optical section or a higher power image with a more restricted field of view.

We agree and remove this statement from the Fig.4 legend.

(4) Some of the graphs are a bit small, recommend reducing the statistical comparison brackets to straight lines, which eliminates a lot of white space and would allow the graphs to be enlarged.

We increased the size of graphs in revised Fig. S2 and Fig.5.

**Reviewer #2 (Significance):**
The authors nicely characterized the role of Ryr in muscle development and function and recreated a human variant of unknown function (RyR1 p.Met4881Ile) in the conserved region of the fly gene. Their data suggested that this variant was likely deleterious as it negatively affected most muscle parameters. This work supports a role for the fly model in testing potential human disease gene variants. The reviewers field of expertise is in Drosophila genetics and in the use of the fly as a model system for understanding the genetic networks contributing to muscle structure and function at the cellular level.
**Reviewer #3:**
SummaryThis paper examines the Drosophila Ryanodine Receptor (RyR or dRyR). Ryanodine receptors are enormous channel proteins that mediate calcium efflux from the endoplasmic reticulum and sarcoplasmic reticulum. One goal of the work is to describe salient developmental features of Drosophila RyR (i.e., where it localizes in the cell and how it contributes to muscle development and function) and to refine knowledge from prior reports. Many of the analyses toward that goal are well done; this reviewer especially likes the examination of how muscles develop (Fig. 5).Another goal is to compare this information with what is known about mammalian RyRs. There seems to be a lot in common between Drosophila and mammalian RyRs. The paper finishes by taking a human ryanodine receptor variant of unknown significance and generating the corresponding amino-acid substitution in Drosophila RyR. The substitution has some phenotypic consequences for fly coordination, so the authors conclude that the human variant is likely to be pathogenic.In terms of investigation, a refined description of RyR biology is welcome. Ryanodine receptors are critical contributors/mediators of intracellular calcium signaling processes. Understanding their properties can help to contextualize the results of studies where calcium dynamics are at play. This is true of for both Drosophila and non-Drosophila work. For this version of the paper, there are several statements that should be edited, both in terms of accuracy and in terms of reporting prior knowledge. Additionally, some experiments are missing controls or reagent verification. Importantly, the anti-RyR antibody needs supporting information regarding its specificity.Main Comments(1) The paper does not fully state what has been done before in terms of studying Drosophila ryanodine receptor expression. In comparing the work on ryanodine receptors in vertebrates versus Drosophila, the authors write, "By contrast, no systematic analyses have yet been performed to assess the expression of the sole Drosophila dRyR gene." I was a little surprised by this sentence, so I examined the literature. There are hundreds of Drosophila publications that mention the ryanodine receptor in some way, but they are not about gene expression . As stated, the sentence might depend on what the authors mean by "systematic analyses." Two early works are relevant here: the Hasan and Rosbash, 1992 paper and the Sullivan et al., 2000 paper. Both are cited in this study. And both of these early papers addressed RyR gene expression, so that fact should be acknowledged up front.

We agree with the Reviewer that there is a large number of publications that mention Drosophila ryanodine receptor with two of them identified by the Reviewer that provide information about Drosophila RyR expression. We refer to both of them and follow Reviewer’s suggestion to further acknowledge their work. The modified sentence in the text reads as follow:

“…in spite of early works by Hasan and Rosbash (1992) and Sullivan et al., (2000) no systematic analyses have yet been performed to assess the developmental expression pattern of the sole Drosophila dRyR gene…”

Concerning “systematic analyses” we mean the analyses of dRyR expression at both transcripts and protein levels during embryonic development and in differentiated muscles.

(2) (Related) I examined those two early papers to cross-check the extent of analysis done previously. The text of Hasan and Rosbash reports in situ examination of RyR transcript using a digoxigenin probe (though the online version of that 1992 paper seems to have left out the relevant mesodermal and muscle images referenced in the paper, in favor of duplicating Figure 5 three times - I emailed Development to alert them). More relevant, several experiments executed in the Sullivan paper agrees closely with the current paper. As such, it needs more complete referencing. The Sullivan paper showed short, round larvae in mutants (Fig. 1 of Sullivan); ubiquitous mRNA, strongly in muscle and mesoderm (Fig. 2 of Sullivan); impaired muscle function in mutants (Fig. 3 of Sullivan), and impaired larval heart rate (Fig. 4 of Sullivan).

Sullivan et al. paper is indeed a reference paper for Drosophila RyR. Our data are however largely novel and/or substantially extending those reported by Sullivan. Notably, we show for the first time developmental dRyR protein expression pattern in embryos and in larval filets, we also analyse dRyR isoform transcripts expression and provide for the first time embryonic muscle phenotype analyses that shed light on so far under investigated developmental function of dRyR.

We follow Reviewer’s suggestion and provide in the revised version additional citations of this work:

“…attenuation of dRyR (C57>dRyR RNAi) led to a significantly reduced larva body length (Fig. 3B, M) compared to control (Fig. 3A, Q), an observation that correlates with previously observed (Sullivan et al., 2000) reduced body size of dRyR^16^ mutant larvae…”.

“…our data extend previous observations of affected muscle contractility in RyR mutants (Sullivan et al., 2000)…”

“…Overall, observed dRyR loss-of-function heart phenotypes with a slow heart rate and increased arrhythmia correlate with impaired cardiac function in RyR mutant larvae (Sullivan et al., 2000)…”

(3) Fig. 1B-D (antibody staining): There are puzzles with this experiment. The first is with the anti-Dlg channel. Dlg is a core component of the NMJ postsynaptic density, and the antibody reveals a bright cage of Dlg around the boutons. But with the muscle images in Figure 1B, there are no boutons apparent (unless they are so far out of focus as to be invisible).

Indeed, Dlg also stains postsynaptic NMJs at the muscle surface. On the Fig. 1B showing more internal optical sections to reveal T tubules Dlg-positive NMJs are out of focus.

The second question centers on the dRyR antibody. The results state, "We first tested the expression of dRYR at the protein level." This sentence appears immediately after the sentence for gene expression from point 1. Technically, this antibody will help determine protein localization, not gene expression. But more importantly, there is no supporting/verifying information about this guinea pig anti-dRYR antibody. The methods state that it was provided by Robert Scott from NIMH. But there is no accompanying citation, no information about the antigen used to raise the antibody, and no negative control (either mutant or RNAi) to show that the staining is specific. If this is a published anti dRyR antibody that already meets the standards of specificity, that should be made clear, and the citation should be given. But if not, the information and data about the production of the antibody and the testing of its quality needs to be shared.

We apologize for this omitted citation. The anti-dRyR antibody has been previously described and its specificity tested in the article Gao et al., (2013). Corresponding author of this paper David J. Sandstrom left NIMH and anti-dRyR antibodies are currently curated by Rob Scott from Benjamin White’s lab at NIMH.

He generously sent us sample of this antibody. We add this information to the Material and Methods section.

(4) Fig. S1: Similar to the antibody, is there a negative control probe that does not reveal this expression pattern? There are any number of probes or secondary antibodies that non-specifically label Drosophila muscles in patterns just like this.

We are confident that the HCR probes are working properly as they reveal dRyR transcripts expression that is consistent with dRyR protein expression pattern. In parallel they show differential expression in embryos.

Author response image 3 shows the control HCR ISH experiment with a probe that detects Apterous transcripts (specific for a subset of embryonic muscles and not present in L3 larval muscles).

**Author response image 3. sa3fig3:** A comparison between Ap HCR (A, A’) and dRyR Ex23 HCR (E, E’) signals.

Minor Comments(1) "Overall, observed dRYR loss-of-function heart phenotypes...are reminiscent of those associated with aging (Nishimura et al., 2010), indicating that dRyR RNAi-induced impairment of Ca2+ homeostasis contributes to cardiac aging..." The conclusion of the sentence does not logically follow from the first part. This is because the tests conducted here were on rhythm, not on calcium homeostasis and cardiac aging.

So, the tests cannot definitively say anything about those latter phenotypes.

To answer this reviewer’s coment we modify the concluding sentence as follow:

“…We hypothesize that dRyR RNAi-induced impairment of Ca2+ homeostasis could contribute to cardiac aging, for which Drosophila is a recognized model (Nishimura et al., 2011).”

(2) Fig. S2 (bar graph): "% of total" - Is this supposed to refer to the percentage of the total muscle area that is positive for ATP5a staining? That should be clarified.

We provide clarification in the Fig.S2 legend. “% of total” means the percentage of the measured muscle area that is positive for ATP5a staining”.

(3) Fig. 3M, should say length

Done

(4) Fig. 5A legend - See Sullivan; that paper concluded that RyR[16] was hypomorphic instead of null, based on RyR[16]/Df comparison to RyR[16]/RyR[16]. Intuitively, I agree; a lesion that rips out the start site would likely be null. The antibody could help with classifying the allele, depending on the part of RyR used as the antigen.

The RyR^16^ mutants were indeed described by Sullivan et al., as hypomorphic and not null. In the Fig. 5 legend we modify the comment to: “…homozygous dRyR^16^ mutant embryo…”

(5) Discussion: "This also suggests that all dRyR isoforms are collectively required for larval muscle function." That sentence does not logically follow the expression information. In order to test that idea, individual isoforms would need to be eliminated or knocked down.

We agree with this comment and modify our sentence accordingly.

“However, whether all dRyR isoforms are collectively required for larval muscle function requires further investigation.”

**Reviewer #3 (Significance):**
The idea that RyR is expressed in many kinds of muscle is put forth as a major conclusion. It is good that the authors report this fact, and the impacts on muscle development documented in Figure 5 are some of the best data in the paper. However, in terms of opening up a new understanding of RyR biology, the impact of this information seems modest. Prior Drosophila work and the work of others studying these channels show that ryanodine receptors are ubiquitous. The fact that there is only one Drosophila RyR gene would lead most scientists to hypothesize that it would be present on the ER surfaces of all kinds of tissues, including different types of muscle.Novel phenotypic information for Drosophila RyR is reported in the study, and this is good. But in terms of the model system, the strength of Drosophila is in using genetic combinations to make refined conclusions. That toolkit is not fully used here; therefore, the paper is mostly descriptive. This study is mostly a single-gene study (dRyR), with isolated exceptions, like Cam knockdown in Figure 5.

To improve the functional/mechanistic aspect of the manuscript in the revised version we include to Fig.5 the analysis of myogenic role of additional calcium regulator: ER calcium pump SERCA.